# Prevalence of bacterial vaginosis and aerobic vaginitis and their associated risk factors among pregnant women from northern Ethiopia: A cross-sectional study

Gebrehiwet Tesfay Yalew[1], Saravanan Muthupandian[2]*, Kiflom Hagos[3], Letemichael Negash[3], Gopinath Venkatraman[4], Yemane Mengsteab Hagos[1], Hadush Negash Meles[1], Hagos Haileslasie Weldehaweriat[1], Hussein O. M. Al-Dahmoshi[5], Morteza Saki[6]*

1 Department of Medical Laboratory Science, College of Medicine and Health Sciences, Adigrat University, Adigrat, Ethiopia, 2 AMR and Nanomedicine Laboratory, Center for Transdisciplinary Research, Department of Pharmacology, Saveetha Dental College, Saveetha Institute of Medical and Technical Sciences (SIMATS), Chennai, India, 3 Department of Microbiology and Immunology, Division of Biomedical Sciences, School of Medicine, College of Health Sciences, Mekelle University, Mekelle, Ethiopia, 4 University of Malaya Proteomics Research Center, Deputy Vice-Cancellor (Research & Innovation), University of Malaya, Kuala Lumpur, Malaysia, 5 Department of Biology, College of Science, University of Babylon, Babylon, Hilla City, Iraq, 6 Department of Microbiology, Faculty of Medicine, Ahvaz Jundishapur University of Medical Sciences, Ahvaz, Iran

* bioinfosaran@gmail.com (SM); mortezasaki1981@gmail.com (MS)

**Data Availability Statement:** All relevant data are within the paper and its Supporting Information files.

## Abstract

This study aimed to determine the prevalence of bacterial vaginosis (BV) and aerobic vaginitis (AV) and their associated risk factors among pregnant women from Ethiopia. Also, this study investigated the bacterial pathogens and their antibiotic resistance in AV cases. A total of 422 pregnant women from northern Ethiopia were participated in this study. Socio-demographic and clinical data were recorded. Vaginal swabs were collected and used for wet mount and Gram stain methods to evaluate the AV and BV scores according to the Nugent's and Donder's criteria, respectively. In AV cases the bacterial pathogens and their antibiotic resistance were determined using standard methods. The possible risk factors for AV and BV in pregnant women were investigated. The prevalence rates of BV and AV were 20.1% (85/422) and 8.1% (34/422), respectively. BV was more common in symptomatic vs. asymptomatic people ($P < 0.001$), and in second trimester vs. first trimester samples ($P = 0.042$). However, AV was more common in secondary school vs. primary and those who were unable to read and write ($P = 0.021$) and in housewife women vs. employee ($P = 0.013$). A total of 44 bacterial strains were isolated from AV cases, of which the coagulase-negative staphylococci (CoNS) (38.6%) and *Staphylococcus aureus* (29.5%) were the most predominant bacteria, respectively. The highest resistance rate was observed against penicillin (100.0%) in staphylococci, while 86.7% of them were sensitive to ciprofloxacin. The resistance rate of *Enterobacteriaceae* ranged from 0.0% for ciprofloxacin and chloramphenicol to 100.0% against amoxicillin/clavulanate. The prevalence of BV was higher than AV in pregnant women. This higher prevalence of BV suggests that measures should be taken to

**Funding:** This research was financed by the Mekelle University, Mekelle -1871, Ethiopia.

**Competing interests:** The authors have declared that no competing interests exist.

reduce the undesired consequences related to BV in the pregnancy. The circulation of drug-resistant bacteria in vaginal infections requires a global surveillance to reduce the risks to pregnant mothers and infants.

## Introduction

Bacterial vaginosis (BV) is an imbalance in the normal vaginal flora with decreased levels of the usual predominant lactobacilli and proliferation of various pathogenic mixed flora of aerobic, anaerobic and microaerophilic species [1, 2]. Aerobic vaginitis (AV), a term coined in a hallmark by Donders et al. [3] to emphasize its difference from BV, is characterized by inflammation of the vaginal epithelium and also characterized by abnormal vaginal microflora containing aerobic and enteric bacteria like *Escherichia coli*, *Klebsiella* spp., *Acinetobacter* spp., *Staphylococcus* spp., *Enterococcus* spp., and group B *Streptococcus* (GBS). However, both AV and BV are vaginal dysbioses characterized by the reduction in lactobacilli [3–7].

It is well known that unfavorable outcomes of early pregnancy including premature rupture of membranes (PROM), chorioamnionitis, preterm delivery, spontaneous abortion, and low birth weight can be exacerbated by BV during pregnancy [8–10]. Likewise, AV without diagnosis or treatment, can cause perinatal complications such as preterm birth, PROM, and fetal infections in pregnant women [11–14]. A critical part of AV is a shift in the vaginal flora from *Lactobacillus*-dominated bacteria to aerobic bacteria, which alters the vaginal microbiome and leads to negative perinatal outcomes. In AV, certain bacteria create sialidase enzyme, which breakdown host defense components like IgA. These enzymes stimulate the release of sialic acid from mucins and mucosal epithelial cells. Further, AV appears to be associated with elevated levels of some cytokines including IL-1b, IL-6, and IL-8, which are known risk factors for negative pregnancy outcomes [11]. Despite the fact that some studies have looked at the link between AV and pregnancy outcomes, AV is still poorly understood and there are few studies that have addressed the microbial diversity of AV in pregnant women [11–14]. AV, even when asymptomatic, can increase the risk of chorioamnionitis, and result in a neonatal mortality rate of 25.0–90.0% due to congenital neonatal sepsis [11].

Epidemiological studies have shown that older maternal age, multiple sexual partners, previous spontaneous miscarriages, and alteration of vaginal bacterial communities are among the risk factors for AV and BV [15]. Women who have a new sexual partner, multiple sexual partners, smoking habits and use of intra-uterine devices also have an increased predisposition to acquire BV [15–19].

A growing body of data suggests that BV facilitates the acquisition of sexually transmitted infections (STIs) such as *Neisseria gonorrhoeae*, *Chlamydia trachomatis*, human immunodeficiency virus (HIV) and herpes simplex virus type-2 infection (HSV-2) [4, 14, 19, 20]. Moreover, the shedding of HSV-2 in the genital tract is significantly higher in women with BV than in BV free women [19]. Despite the availability of effective treatment regimens, recurrence is common and can cause significant frustration in women with BV. Approximately 84.0% of women with BV are asymptomatic [16, 21].

Worldwide, almost one-third of women are positive for BV and a higher prevalence has been found in pregnant women from developing countries (Latin America, Asia, and most African countries), with the highest prevalence of BV reported from several African countries [22–24]. However, the prevalence of AV ranges from 7% to 12% but during pregnancy, it seems to be lower, in the range of 4.0 to 8.0% [3]. Previous studies have identified the

unmarried status, frequent vaginal douching, and long-term use of pregnancy-preventing drugs, and previous history of vaginal infection as the associated risk factors for AV [25, 26].

In Ethiopia, different studies reported that the prevalence of BV ranges from 0.5% to 48.6% [4, 8, 27] but among pregnant women the range was from 0.5% to 19.4% [8, 27]. Although the prevalence of BV has been reported from different parts of Ethiopia as well as other developing countries [4, 8, 27–29], to our knowledge, there has been no study of AV in Ethiopia. Moreover, the studies conducted in Ethiopia on BV did not also cover North Ethiopia especially Mekelle city and local data regarding BV are sparse in Ethiopia [4, 8, 27]. Hence, this study aimed to determine the prevalence of BV, AV, and their associated risk factors among pregnant women attending antenatal care clinics in Ayder Comprehensive Specialized Hospital (ACSH), Tigray, Northern Ethiopia. Also, the prevalence of bacterial strains isolated from AV cases and their antibiotic susceptibility patterns were investigated.

## Materials and methods

### Ethical clearance

Ethical clearance was obtained from Mekelle University, College of Health Science Ethical Review Committee (ERC 1211/2019) in accordance with the Deceleration of Helsinki. Permission letter was secured from the ACSH and written informed consent was obtained from the study participants before proceeding to data collection. Confidentiality of the result was also maintained.

### Study design, period and area

This hospital-based cross-sectional study was conducted from February to June 2019 at Ayder Comprehensive Specialized Hospital (ACSH), Mekelle city, Northern Ethiopia. ACSH is a University affiliated hospital with a total capacity of about 500 inpatient beds in all departments and other specialty units including antenatal clinics and gynecology unit. The hospital serves 9 million referral and non-referral patients from all parts of the Tigray region and other neighborhood regions such as Afar and Amhara regional states including the Eritrean refugees.

### Studied population

The sample size of this study was determined using the convenience sampling as a pilot design. All symptomatic and asymptomatic out-patient pregnant women who visited the obstetrical and gynecological clinic for antenatal care services at the ACSH were included consecutively in the study, completed the questionnaire adequately and provided vaginal swab specimens. Socio-demographic data (age, marital status, educational) and clinical findings: abnormal vaginal discharge or fluid, and sexual and behavioral characteristics: vaginal hygiene, number of lifetime sexual partners were obtained using interview based structural questionnaire by trained midwives and supervision by the principal investigator. Additionally, some independent variables such as HIV and syphilis were recorded from the registration book.

### Exclusion criteria

Pregnant women who have taken antibiotics in the preceding two weeks of data collection, those with unknown source of vaginal bleeding, women under legal age, those with genital malignancy and those who douched their vagina with chemicals were excluded from the study.

## Specimen collection, handling, and transportation

Upon admission to the study, the gynecologist/physician performed a clinical examination of each participant and recorded signs of vaginal discharge and other complications. During the examination, two vaginal swabs were collected aseptically from posterior vaginal fornix/ lateral wall of the vagina using sterile rayon tipped applicator stick and dipped into a sterile tube containing two drops of sterile physiological saline and taken to the laboratory within 30 minutes of collection for laboratory processing. One swab was used for both wet mount and smeared on clear slide for BV and AV scores. The other applicator stick was used for culture if AV score was positive.

## Wet mount examination

To evaluate the presence of clue cells (epithelial cells with hazy borders due to the attendance of bacteria) for BV, parabasal cells (sign of severe epithelial inflammation), better distinction between toxic and normal leukocytes, and recognition of lactobacillary grades for AV, the wet mount was prepared from vaginal discharge on the slide examined by the high power (400x) of the microscope [9, 30].

## Gram staining for BV and AV scores

The Gram-stained smears were prepared from vaginal discharges. The diagnosis of BV was performed according to the Nugent et al. [31] scoring system. The Gram-stained smears were evaluated for the following morphotypes under oil immersion (1000x magnification): large Gram-positive rods (*Lactobacillus* spp.), small Gram-variable rods (*Gardnerella* spp.), small Gram-negative rods (*Bacteroides* spp.), and curved Gram-variable rods (*Mobiluncus* spp.). Each morphotype was quantitated from 1 to 4+ with regard to the number of morphotypes per oil immersion field (0, no morphotypes; 1+, less than 1 morphotypes; 2+, 1 to 4 morphotypes; 3+, 5 to 30 morphotypes; 4+, 30 or more morphotypes). This scale goes the opposite direction for *Lactobacillus* morphotypes i.e. no morphotypes = 4. The normal score is represented by values between 0 and 3, while values between 4 and 6 represent an intermediate vaginal microbiota, and finally values between 7 and 10 have been considered diagnostic for BV [31].

AV diagnosis was done using Gram staining under dry high power objective (400x) (to determine AV score) and oil immersion magnification (for identification of organisms). AV score was calculated by determining the presence or absence of lactobacilli, type of vaginal flora, the number of leukocytes, and parabasal epithelial cells using 400x magnification, according to a modified Donder's score [7]. An AV score of less than 3 was defined as normal AV, 3 to 4 as light AV, 5 to 6 as moderate AV, and any score > 6 as severe AV [3, 7].

## Bacterial isolates

All vaginal swabs with positive AV score were plated on to 5.0% sheep blood agar, MacConkey agar, mannitol salt agar, and chocolate agar (Oxoid, Basingstoke, UK) to isolate aerobic bacteria. The inoculated media incubated at 37°C aerobically for 18–24 hours [7, 32]. The pure isolates of the bacterial pathogen were primarily characterized by colony morphology, hemolytic reactions on blood agar plates and Gram stain. Identification of aerobic bacteria to genus and/ or species level was done using a series of routine biochemical tests such as catalase, coagulase, indole production, gas production, urease, $H_2S$ production, citrate utilization, motility, and fermentation of various carbohydrates [33].

## Antibiotic susceptibility testing

Antibiotic susceptibility testing (AST) was performed using the modified Kirby-Bauer disc diffusion method according to the clinical laboratory standard Institute (CLSI) guidelines [34]. The following antimicrobial drugs were employed for Gram-positive bacteria (GPB): penicillin (10 μg), trimethoprim/sulfamethoxazole (1.25/23.75 μg), clindamycin (2 μg), erythromycin (15 μg), gentamycin (10 μg), ciprofloxacin (5 μg), tetracycline (30 μg), doxycycline (30 μg) and chloramphenicol (30 μg) (Oxoid, England) and for Gram-negative bacteria (GNB): amikacin (30 μg), tobramycin (10 μg), ampicillin (10 μg), amoxicillin/clavulanate (30 μg), meropenem (10 μg), gentamycin (10 μg), ciprofloxacin (5 μg), tetracycline (30 μg), doxycycline (30 μg) and chloramphenicol (30 μg) (Oxoid, England). The sensitivity test results were interpreted according to the CLSI 2018 [34]. Reference strains from Ethiopian Public Health Institute, *E. coli* ATCC 25922 and *S. aureus* ATCC 25923 were used for quality control.

## Data processing and analysis

Collected quantitative data were coded and analyzed using Statistical Package for social sciences (SPSS), version 22. The frequency and percentage of each variable were calculated using cross-tabulations. Statistical analysis like logistic regression for odds ratios at 95% confidence interval (CI), univariate, and multivariate analysis were performed to calculate the association of selected exposure variables with the outcome variable and to check the association of possible risk factors with BV and AV in pregnant women. An a priori selected set of variables with a *P*-value < 0.2 in the univariate analysis were considered for the multivariate regression analysis. *P*-value < 0.05 was considered as statistically significant.

## Results

### Overall socio-demographic characteristics

A total of 422 pregnant women were examined for their vaginal microbiota in ACSH. The majority of the study participants were in the age group of 21–29 years (64.0%), followed by ≥ 30 years (27.3%), and ≤ 20 years (8.7%), respectively. The mean ± SD of the participants' age was 27.2 ± 5.0 years (range of 18–47 years). Most of the study participants were married 402 (95.3%) and urban residents 400 (94.8%). In addition, the majority of the study participants had completed secondary school 171 (40.5%) and were housewives 223 (52.8%) (Table 1).

### Total prevalence of vaginal infections among pregnant women

The overall prevalence of different vaginal infections was 27.7% (117/422). The total prevalence of BV was 20.1% (85/422) which included BV alone 12.3% (52/422), BV with AV 5.5% (23/422), BV with candidiasis 1.7% (7/422), and BV + AV + trichomoniasis 0.7% (3/422). The total prevalence of AV was 8.1% (34/422) which included AV alone 1.4% (6/422), AV with BV 5.5% (23/422), AV with candidiasis 0.2% (1/422), AV + BV + trichomoniasis 0.7% (3/422), and AV with trichomoniasis 0.2% (1/422). The total prevalence of trichomoniasis (*Trichomonas vaginalis*), candidiasis, and mixed infections were 2.1% (9/422), 6.4% (27/422), and 8.3% (35/422), respectively (Table 2). Among the mixed infections, the majorities 65.7% (23/35) had mixed BV + AV followed by mixed BV + candidiasis 20.0% (7/35) (Table 2). In total, 77.3% (326/422) had a normal score (BV score of 0–3 and AV score of 0–2). Also, 62.3% (263/422), 20.1% (85/422), and 17.5% (74/422) had a normal BV score, definitive BV, and intermediate vaginal microbiota, respectively. Similarly, most of the study participants 91.9% (388/422) had a normal AV score. Among the pregnant women, 4.5% (19/422), 0.9% (4/422), and 2.6% (11/

**Table 1. Socio-demographic characteristics of pregnant women at Ayder comprehensive Specialized Hospital from February to June 2019.**

| Variables | | Count | Percent |
|---|---|---|---|
| Age | ≤ 20 years | 37 | 8.7 |
| | 21–29 years | 270 | 64.0 |
| | ≥ 30 years | 115 | 27.3 |
| Residence | Urban | 400 | 94.8 |
| | Rural | 22 | 5.2 |
| Educational status | Unable to write and read | 19 | 4.5 |
| | Primary school | 82 | 19.4 |
| | Secondary school | 171 | 40.5 |
| | College and above | 150 | 35.5 |
| Occupational status | Employee | 111 | 26.3 |
| | Housewife | 223 | 52.8 |
| | Others | 88 | 20.9 |
| Marital status | Unmarried | 15 | 3.6 |
| | Married | 402 | 95.3 |
| | Divorced/widowed | 5 | 1.1 |
| HIV | Positive | 12 | 2.8 |
| | Negative | 410 | 97.2 |
| Condom use | Yes | 31 | 7.3 |
| | No | 391 | 92.7 |
| Fungal infection | Yes | 33 | 7.8 |
| | No | 389 | 92.2 |
| Number of LTSP | One | 363 | 86.0 |
| | Two and above | 59 | 14.0 |
| Number of pantyliner used per day | 1-2/day | 298 | 70.6 |
| | 1/2-4 days | 124 | 29.4 |
| Douching using water | Once daily | 119 | 28.2 |
| | More than one per day | 303 | 71.8 |
| Douching using soap | Yes | 43 | 10.2 |
| | No douching | 379 | 89.8 |
| Previous BV/GTI | Yes | 62 | 14.7 |
| | No | 360 | 85.3 |
| History of abortion | Spontaneously | 64 | 15.2 |
| | Induced | 32 | 7.6 |
| | No | 326 | 77.3 |
| Gestational age | First trimester | 51 | 12.1 |
| | Second trimester | 207 | 49.1 |
| | Third trimester | 164 | 38.9 |
| Number of pregnancy | Primigravida | 158 | 37.4 |
| | Multigravida | 264 | 62.6 |
| BV score | Normal | 263 | 62.3 |
| | Intermediate | 74 | 17.5 |
| | BV | 85 | 20.1 |
| AV score | Normal | 388 | 91.9 |
| | Light | 19 | 4.5 |
| | Moderate | 4 | 0.9 |
| | Severe | 11 | 2.6 |

HIV = Human immunodeficiency virus, LTSP = Lifetime sexual partner, AV = Aerobic vaginitis, BV = Bacterial vaginosis, GTI = Genital tract infection

**Table 2. Prevalence of vaginal infections among pregnant women at Ayder Comprehensive Specialized Hospital from February to June 2019.**

| Vaginal infections | Frequency | Percent |
|---|---|---|
| Bacterial vaginosis (BV) | 52 | 12.3 |
| Aerobic vaginitis (AV) | 6 | 1.4 |
| Candidiasis | 19 | 4.5 |
| Trichomoniasis | 5 | 1.2 |
| Total | 82 | 19.2 |
| **Mixed infection** | | |
| BV + AV | 23 | 5.5 |
| BV + candidiasis | 7 | 1.7 |
| BV + AV + trichomoniasis | 3 | 0.7 |
| AV + candidiasis | 1 | 0.2 |
| AV + trichomoniasis | 1 | 0.2 |
| Total | 35 | 8.3 |
| Normal women without any vaginal infections | 305 | 72.3 |
| Women with normal scores for BV and AV | 326 | 77.3 |
| Total | 422 | 100 |

BV = Nugent 7–10, AV = score > 2, Candidiasis and trichomoniasis = wet mount +.

422) had light, moderate, and severe AV, respectively (Table 1). The prevalence of BV among symptomatic pregnant women was 35.1% (n = 27/77), but among asymptomatic, the prevalence was 16.8% (n = 58/345). The majority (68.2%, n = 58/85) of BV diagnosed pregnant women were asymptomatic. Statistical analysis showed that symptomatic pregnant women were 2.7 times higher to be positive for BV than asymptomatic women and BV is significantly associated with symptoms of white homogenous discharge ($P = 0.001$). However, symptoms of AV were not significantly associated with AV positive results ($P = 0.549$). The prevalence of AV among symptomatic pregnant was 8.8% (n = 3/27), whereas 7.8% (n = 31/ 395) of AV positive participants were from asymptomatic pregnant women.

## Socio-demographic characteristics of pregnant women with bacterial vaginosis

The socio-demographic characteristics of pregnant women with BV are presented in Table 3. The prevalences of BV among age categories were similar (21.6, 20.7 and 18.3%) and without statistical differences. Statistical analysis showed that there was no significant association between BV and age, place of residence, educational status, marital status, occupation, HIV, abortion history, previous BV/genital tract infection (GTI), vaginal bathing, number of pantyliner used, and the other studied factors ($P$-value was > 0.05) (Table 3).

However, pregnant women sampled in the second trimester had a significantly higher prevalence of BV (23.7%) than those who were in the first and third trimester with a prevalence of 17.6% and 16.5%, respectively ($P = 0.042$). In bivariate analysis, symptoms of white homogenous discharge, educational status, gestational age, and occupational status showed significant association with BV ($P$-value < 0.05). However, after adjustment for confounders in multivariate analyses (Table 3); only symptoms of BV [2.672 (1.547, 4.615), ($P < 0.001$)] and second trimester [0.563 (0.324, 0.979), ($P = .042$)] were found significantly associated with BV ($P < 0.05$).

**Table 3. Univariate and multivariate analysis of factors associated with bacterial vaginosis among pregnant women attending antenatal care in Ayder Comprehensive Specialized Hospital from February to June 2019.**

| Variables | | BV positive n (%) | BV negative n (%) | Univariate | | Multivariate | |
|---|---|---|---|---|---|---|---|
| | | | | COR (95% CI) | *P*-value | AOR (95% CI) | *P*-value |
| Age(year) | ≤ 20 | 8 (21.6) | 29 (78.4) | 0.810 (0.324, 2.021) | 0.651 | | |
| | 21–29 | 56 (20.7) | 214 (79.3) | 0.854 (0.489,1.490) | 0.578 | | |
| | ≥ 30 | 21 (18.3) | 94 (81.7) | 1 | | | |
| Place of residence | Urban | 80 (20.0) | 320 (80.0) | 1.176 (0.421, 3.285) | 0.756 | | |
| | Rural | 5 (22.7) | 17 (77.3) | 1 | | | |
| Educational status | Unable to read and write | 4 (21.0) | 15(79.0) | 0.645 (0.196, 2.123) | 0.470 | 0.734 (0.207, 2.605) | 0.632 |
| | Primary | 17 (20.7) | 65 (79.3) | 0.657 (0.326, 1.323) | 0.240 | 0.866 (0.389, 1.930) | 0.725 |
| | Secondary | 42 (24.6) | 129 (75.4) | 0.528 (0.298, 0.934) | 0.028* | 0.645 (0.338, 1.228) | 0.182 |
| | College and above | 22 (14.7) | 128 (85.3) | 1 | | | |
| Occupation | Employee | 16 (14.4) | 95 (85.6) | 1.979 (0.967, 4.052) | 0.062 | 1.445 (0.650, 3.214) | 0.366 |
| | Housewife | 47 (21.1) | 176 (78.9) | 1.248 (0.699, 2.229) | 0.454 | 1.081(0.579, 2.020) | 0.619 |
| | Others | 22 (25.0) | 66 (75.0) | 1 | | | |
| Marital status | Unmarried | 5 (33.3) | 10 (66.7) | 1 | | | |
| | Married | 78 (19.4) | 324 (80.6) | 2.077 (0.690, 6.249) | 0.193 | 1.557 (0.470. 5.155) | 0.469 |
| | Divorced/widowed | 2 (25.0) | 3 (75.0) | 0.750 (0.093, 6.043) | 0.787 | 2.971 (0.404. 21.838) | 0.285 |
| Cigarette smoking | Yes | 0 (0.0) | 1 (100.0) | - | - | | |
| | No | 85 (20.2) | 336 (79.8) | 1 | | | |
| HIV | Positive | 3 (25.0) | 9 (75.0) | 0.750 (0.199, 2.833) | 0.671 | | |
| | Negative | 82 (20.0) | 328 (80.0) | 1 | | | |
| Syphilis | Positive | 1 (33.3) | 2 (66.7) | 0.501 (0.045, 5.597) | 0.575 | | |
| | Negative | 84 (20.0) | 335 (80.0) | 1 | | | |
| Condom use | Yes | 6 (19.4) | 25 (80.6) | 1.055 (0.419, 2.659) | 0.910 | | |
| | No | 79 (20.2) | 312 (79.8) | 1 | | | |
| Previous fungal infection | Yes | 8 (24.2) | 25 (75.8) | 0.771 (0.335, 1.776) | 0.542 | | |
| | No | 77 (19.8) | 312 (80.2) | 1 | | | |
| Number of LTSP | One | 72 (19.8) | 291 (80.2) | 1 | | | |
| | Two and above | 13 (22.0) | 46 (78.0) | 0.875 (0.449, 1.707) | 0.696 | | |
| Number of pantyliner used/day | 1-2/day | 63 (21.1) | 235 (78.9) | 1 | | | |
| | 1/2-4 days | 22 (17.7) | 102 (82.3) | 1.243 (0.711, 2.089) | 0.428 | | |
| Douching using water | Once daily | 27 (22.7) | 92 (77.3) | 1 | | | |
| | More than once daily | 58 (19.1) | 245 (80.9) | 1.240 (0.740, 2.076) | 0.414 | | |

(*Continued*)

**Table 3.** (Continued)

| Variables | | BV positive n (%) | BV negative n (%) | Univariate | | Multivariate | |
|---|---|---|---|---|---|---|---|
| | | | | COR (95% CI) | *P*-value | AOR (95% CI) | *P*-value |
| Douching using soap | Yes | 8 (18.6) | 35 (81.4) | 1.115 (0.497, 2.502) | 0.791 | | |
| | No | 77 (20.3) | 302 (79.7) | 1 | | | |
| Previous BV/GTI | Yes | 11 (17.7) | 51 (82.3) | 1.200 (0.596, 2.416) | 0.610 | | |
| | No | 74 (20.6) | 286 (79.4) | 1 | | | |
| Previous history of abortion | Once | 15 (23.4) | 49 (76.6) | 0.783 (0.412, 1.485) | 0.453 | | |
| | Spontaneously | 7 (21.9) | 25 (78.1) | 0.856 (0.354, 2.067) | 0.729 | | |
| | No | 63 (19.3) | 263 (80.7) | 1 | | | |
| Number of the sexual partner in the last 12 months | One | 85 (20.2) | 336 (79.8) | 1 | | | |
| | More than two | 0(0) | 1(100.0) | - | - | | |
| Gestational age | 1st trimester | 9 (17.6) | 42 (82.4) | 0.920 (0.401, 2.109) | 0.843 | 0.843 (0.359, 1.980) | 0.695 |
| | 2nd trimester | 49 (23.7) | 158 (76.3) | 0.635 (0.377, 1.072) | 0.089 | 0.563 (0.324, 0.979) | 0.042* |
| | 3rd trimester | 27 (16.5) | 137 (83.5) | 1 | | | |
| Number of pregnancy | Primigravida | 28 (17.7) | 130 (82.3) | 1 | | | |
| | Multigravida | 57 (21.6) | 207 (78.4) | 1.278 (0.773, 2.114) | 0.334 | | |

COR = Crude odd ratio, AOR = Adjusted odd ratio, CI = confidence interval, ANC = Antenatal care, HIV = Human immunodeficiency virus, LTSP = Lifetime sexual partner, BV = Bacterial vaginosis, GTI = Genital Tract Infection

* = Significant association

## Socio-demographic characteristics of pregnant women with aerobic vaginitis

As shown in Table 4, the prevalence of AV varied with different socio-demographic characteristics and behavioral factors. Pregnant women aged 30 years and above had a higher prevalence (9.6%) of AV than women aged 21–29 years and those lower than 20 years of age in which the prevalence was 7.4% and 8.1%, respectively. However, statistical analysis showed that there was no significant association between AV and age. In this study, higher prevalence of AV was seen among urban resident pregnant women (8.1%), women with secondary school education (12.3%), unmarried and divorced/widowed women (20%), HIV-positive women (16.7%), condom users (9.7%), women with previous fungal infection (15.2%), women with more than two sexual partner (10.2%), women with no previous BV/GTI cases (8.3%), and women with spontaneous abortion (12.5%). Vaginal douching and number of pantyliners used per day were not associated with AV risk. Pregnant women with the second trimester had a higher prevalence (8.7%) than those who were in the first and third trimester with a prevalence of (7.8%) and (7.3%), respectively. In addition, pregnant women who were pregnant for the first time (primigravida) had less prevalence of AV (7.0%) as compared to those who had been pregnant before (multigravida) (8.7%), but the number of pregnancy was not significantly associated with AV (Table 4). In the bivariate analysis, educational status, occupation and previous fungal infection showed significant association with AV. However, after adjustment of confounders in multivariate analyses, secondary school vs. primary school and those who were unable to read and write [0.292 (0.102, 0.833)], *P* = 0.021] and housewife vs. employee [2.856 (1.250, 6.523),

**Table 4. Univariate and multivariate analysis of factors associated with aerobic vaginitis among pregnant women attending antenatal care in Ayder Comprehensive Specialized Hospital from February to June 2019.**

| Variables | | AV positive n (%) | AV negative n (%) | Univariate analysis | | Multivariate analysis | |
|---|---|---|---|---|---|---|---|
| | | | | COR (95% CI) | P-value | AOR (95% CI) | P-value |
| Age (year) | ≤ 20 | 3 (8.1) | 34 (91.9) | 1 | | | |
| | 21–29 | 20 (7.4) | 250 (92.6) | 1.103 (0.311, 3.909) | 0.879 | | |
| | ≥ 30 | 11 (9.6) | 104 (91.4) | 0.834 (0.220, 3.167) | 0.790 | | |
| Residence | Urban | 34 (8.5) | 366 (91.5) | 1 | 0.240 | | |
| | Rural | 0 (0) | 22 (100) | - | | | |
| Educational status | Unable to read and write | 1 (5.3) | 18 (94.7) | 0.750 (0.085, 6.588) | 0.795 | 0.671(0.064, 7.038) | 0.739 |
| | Primary | 6 (7.3) | 76 (92.7) | 0.528 (0.165, 1.692) | 0.282 | 0.499 (0.133, 1.873) | 0.303 |
| | Secondary | 21 (12.3) | 150 (87.7) | 0.298 (0.117, .759) | 0.011* | 0.292 (0.102, 0.833) | 0.021* |
| | College and above | 6 (4.0) | 144 (96.0) | 1 | | | |
| Occupation | Employee | 6 (5.4) | 105 (94.6) | 3.311(1.216, 9.014) | 0.019* | 2.003 (0.654, 6.141) | 0.224 |
| | Housewife | 14 (6.3) | 209 (93.7) | 2.824 (1.286, 6.203) | 0.010* | 2.856 (1.250, 6.523) | 0.013* |
| | Others | 14 (15.9) | 74 (84.1) | 1 | | | |
| Marital status | Unmarried | 3 (20.0) | 12 (80.0) | 1 | | | |
| | Married | 30 (7.5) | 372 (92.5) | 3.100 (.829, 11.590) | 0.093 | 3.182 (.771. 13.121) | 0.109 |
| | Divorced/widowed | 1 (20.0) | 4 (80.0) | 1.000 (.080, 12.557) | 1.000 | 1.998 (.128, 31.175) | 0.622 |
| Cigarette smoking | Yes | 0 (0) | 1 (100.0) | - | | | |
| | No | 34 (8.1) | 387 (91.9) | 1 | | | |
| HIV | Positive | 2 (16.7) | 10 (83.3) | 0.423 (0.089, 2.015) | 0.280 | | |
| | Negative | 32 (7.8) | 378 (92.2) | 1 | | | |
| Syphilis | Positive | 0 (0) | 3 (100.0) | - | | - | |
| | Negative | 34 (8.1) | 385 (91.9) | 1 | | | |
| Condom use | Yes | 3 (9.7) | 28 (90.3) | 1 | | | |
| | No | 31 (7.9) | 360 (92.1) | 1.244 (0.358, 4.325) | 0.731 | | |
| Previous fungal infection | Yes | 5 (15.2) | 28 (84.8) | 0.451(0.162, 1.256) | 0.128 | 0.444(0.152, 1.294) | 0.1377 |
| | No | 29 (7.5) | 360 (92.5) | 1 | | | |
| Number of LTSP | One | 28 (7.7) | 335 (92.3) | 1 | | | |
| | Two and above | 6 (10.2) | 53 (89.8) | 0.738 (0.292, 1.868) | 0.522 | | |
| Number of pantyliner used/day | 1-2/day | 26 (8.7) | 272 (91.3) | 1 | | | |
| | 1/2-4 days | 8 (6.5) | 116 (93.5) | 1.386 (0.609, 3.152) | 0.436 | | |
| Douching using water | Once daily | 7(5.9) | 112 (94.1) | 1 | | | |
| | More than once daily | 27(8.9) | 276 (91.1) | 0.639 (0.270, 1.510) | 0.307 | | |
| Douching using soap | Yes | 2 (4.7) | 41 (95.3) | 1.890 (0.437, 8.179) | 0.394 | | |
| | No | 32 (8.4) | 347 (91.6) | 1 | | | |

(*Continued*)

**Table 4.** (Continued)

| Variables | | AV positive n (%) | AV negative n (%) | Univariate analysis | | Multivariate analysis | |
|---|---|---|---|---|---|---|---|
| | | | | COR (95% CI) | P-value | AOR (95% CI) | P-value |
| Previous BV/GTI | Yes | 4 (6.5) | 58 (93.5) | 1.318 (0.448, 3.881) | 0.616 | | |
| | No | 30 (8.3) | 330 (91.7) | 1 | | | |
| Previous history of abortion | Once | 6 (8.4) | 58 (91.6) | 0.768 (.301, 1.962) | 0.582 | | |
| | Spontaneously | 4 (12.5) | 28 (87.5) | 0.556 (0.180, 1.717) | 0.308 | | |
| | No | 24 (7.4) | 302 (92.6) | 1 | | | |
| Number of the sexual partner in the last 12 months | One | 34 (8.1) | 387 (91.9) | 1 | | | |
| | More than two | 0 (0) | 1 (100.0) | - | | | |
| Gestational age | 1st trimester | 4 (7.8) | 47 (92.2) | 0.928 (0.286, 3.013) | 0.901 | | |
| | 2nd trimester | 18 (8.7) | 189 (91.3) | 0.829 (0.387, 1.774) | 0.629 | | |
| | 3rd trimester | 12 (7.3) | 152 (92.7) | 1 | | | |
| Number of pregnancy | Primigravida | 11 (7.0)) | 147 (93.0) | 1 | | | |
| | Multigravida | 23 (8.7) | 241 (91.3) | 0.784 (0.371, 1.655) | 0.523 | | |

COR = Crude odds ratio, AOR = Adjusted odds ratio, CI = confidence interval, ANC = Antenatal care, HIV = Human immunodeficiency virus, LTSP = Lifetime sexual partner, BV = Bacterial vaginosis, GTI = Genital tract infection, AV = Aerobic vaginitis

* = Significant association

$P$ = 0.013] pregnant women remained independently associated with a decreased (preventive) and increased likelihood of AV positive, respectively.

## Bacterial isolates and their antibiotic susceptibility patterns among pregnant women with aerobic vaginitis

A total of 44 bacterial isolates were recovered from 34 pregnant women with AV, of which 30 (68.2%) and 14 (31.8%) isolates were different *Staphylococcus* and *Enterobacteriaceae* species, respectively. Among the staphylococci, coagulase-negative staphylococci (CoNS) (38.6%, n = 17/44) and *S. aureus* (29.5%, n = 13/44) were the first and the second predominant bacteria, respectively. *E. coli* (25.0%, n = 11/44) was the most predominant *Enterobacteriaceae* followed by *Citrobacter* spp. (4.5%, n = 2/44) and *Klebsiella pneumoniae* (2.3%, n = 1/44).

Table 5 summarizes the overall drug susceptibility profile of *Staphylococcus* strains against nine antibacterial drugs tested. The highest overall resistance rate was observed against penicillin (100.0%), followed by erythromycin (60.0%), trimethoprim/sulfamethoxazole (53.4%), and tetracycline (30%), while 86.7% of the isolates were sensitive to ciprofloxacin followed by gentamicin (83.3%).

The overall antibiotic susceptibility profile of *Enterobacteriaceae* isolates against the eleven antibacterial drugs tested is summarized in Table 6. Amoxicillin/clavulanate had the highest overall resistance rate (100%) against *Enterobacteriaceae* isolates followed by ampicillin (92.9%) and tobramycin (42.8%). All *Enterobacteriaceae* isolates showed 100.0% sensitivity towards ciprofloxacin and chloramphenicol followed by gentamycin (92.9%) and meropenem (85.7%). All the bacterial species isolated were resistant to one or more antimicrobial agents or classes.

**Table 5. Percentage of antibacterial susceptibility pattern of all *Staphylococcus* isolates (n = 30) from Ayder Comprehensive Specialized Hospital from February to June 2019.**

| *Staphylococcus* strains (n) | Pattern | Antibacterial drugs | | | | | | | | |
|---|---|---|---|---|---|---|---|---|---|---|
| | | **PEN (%)** | **CIP (%)** | **DA (%)** | **E (%)** | **CN (%)** | **TE (%)** | **CAF (%)** | **DOX (%)** | **SXT (%)** |
| CoNS (17) | S | 0 (0.0) | 14 (82.4) | 13 (76.6) | 4 (23.6) | 15 (88.2) | 7 (41.2) | 13 (76.5) | 11 (64.7) | 7 (41.2) |
| | I | 0 (0.0) | 0 (0.0) | 2 (11.7) | 2 (11.8) | 1 (5.9) | 5 (29.4) | 1 (5.9) | 2 (11.8) | 1 (5.9) |
| | R | 17 (100) | 3 (17.6) | 2 (11.7) | 11 (64.6) | 1 (5.9) | 5 (29.4) | 3 (17.6) | 4 (23.5) | 9 (52.9) |
| *S. aureus* (13) | S | 0 (0.0) | 12 (92.3) | 9 (69.2) | 0 (0.0) | 10 (76.9) | 6 (46.2) | 11 (84.6) | 8 (61.5) | 6 (46.2) |
| | I | 0 (0.0) | 1 (7.7) | 2 (15.4) | 6 (46.2) | 1 (7.7) | 3 (23.0) | 0 (0.0) | 2 (15.4) | 0 (0.0) |
| | R | 13 (100) | 0 (0.0) | 2 (15.4) | 7 (53.8) | 2 (15.4) | 4 (30.8) | 2 (15.4) | 3 (23.1) | 7 (53.8) |
| Total (30) | S | 0 (0.0) | 26 (86.7) | 22 (73.4) | 4 (13.3) | 25 (83.3) | 13 (43.3) | 24 (80.0) | 19 (63.4) | 13 (43.3) |
| | I | 0 (0.0) | 1 (3.3) | 4 (13.3) | 8 (26.6) | 2 (6.7) | 8 (26.7) | 1 (3.3) | 4 (13.3) | 1 (3.3) |
| | R | 30 (100.0) | 3 (10.0) | 4 (13.3) | 18 (60.0) | 3 (10.0) | 9 (30.0) | 5 (16.7) | 7 (23.3) | 16 (53.4) |

PEN = Penicillin, CIP = Ciprofloxacin, DA = Clindamycin, E = Erythromycin, CN = Gentamycin, TE = Tetracycline, CAF = Chloramphenicol, DOX = Doxycycline, SXT = Trimethoprim/sulfamethoxazole, S = Sensitive, I = Intermediate, R = Resistant

The overall multiple drug resistance (resistance to three or more antimicrobial classes) was 45.5%. Multiple drug resistance was higher among *Staphylococcus* strains (57.1%) than *Enterobacteriaceae* isolates (40.0%). Also, all *E. coli*, *Citrobacter* spp., and *K. pneumoniae* isolates were multiple drug resistant. Meanwhile, CoNS showed a higher multiple drug resistance pattern (62.5%) than *S. aureus* (30.8%) (Table 7).

## Discussion

In the present study, the overall prevalence of BV by Gram stain Nugent scoring criteria was 20.1% (85/422), which was in the range of local studies reported before from Ethiopia that showed prevalence rates of 2.8% to 48.6% [4, 8]. Also, the current prevalence rate was comparable with studies done in Brazil (20.7%), India (19.6% and 21.0%), and Pakistan (21.0%)

**Table 6. Percentage of antibacterial susceptibility pattern of all *Enterobacteriaceae* isolates (n = 14) from the Ayder Comprehensive Specialized Hospital from February to June 2019.**

| *Enterobacteriaceae* (n) | Pattern | Antibacterial drugs | | | | | | | | | | |
|---|---|---|---|---|---|---|---|---|---|---|---|---|
| | | **MER (%)** | **CIP (%)** | **TOB (%)** | **AK (%)** | **AMC (%)** | **CN (%)** | **TE (%)** | **CAF (%)** | **AM (%)** | **DOX (%)** | **SXT (%)** |
| *E. coli* (11) | S | 10 (90.9) | 11 (100.0) | 6 (54.5) | 6 (54.5) | 0 (0.0) | 10 (90.9) | 7 (63.6) | 11 (100.0) | 0 (0.0) | 7 (63.6) | 11 (100.0) |
| | I | 1 (9.1) | 0 (0.0) | 2 (18.2) | 1 (9.1) | 0 (0.0) | 1(9.1) | 2 (18.2) | 0 (0.0) | 1 (9.1) | 2 (18.2) | 0 (0.0) |
| | R | 0 (0.0) | 0 (0.0) | 3 (27.3) | 4 (36.4) | 11 (100) | 0 (0.0) | 2 (18.2) | 0(0.0) | 10 (90.1) | 2 (18.2) | 0 (0.0) |
| *Citrobacter* spp. (2) | S | 1(50.0) | 2 (100.0) | 0 (0.0) | 2 (100) | 0 (0.0) | 2 (100.0) | 1 (50.0) | 2 (100.0) | 0 (0.0) | 0 (0.0) | 0 (0.0) |
| | I | 1(50.0) | 0 (0.0) | 0 (0.0) | 0 (0.0) | 0 (0.0) | 0 (0.0) | 0 (0.0) | 0 (0.0) | 0 (0.0) | 0 (0.0) | 0 (0.0) |
| | R | 0 (0.0) | 0 (0.0) | 2 (100) | 0 (0.0) | 2 (100.0) | 0 (0.0) | 1 (50.0) | 0 (0.0) | 2 (100) | 2 (100.0) | 2 (100.0) |
| *K. pneumoniae* (1) | S | 1 (100.0) | 1 (100.0) | 0 (0.0) | 1 (100.0) | 0 (0.0) | 1 (100.0) | 1 (100) | 1 (100) | 0 (0.0) | 1 (100.0) | 0 (0.0) |
| | I | 0 (0.0) | 0 (0.0) | 0 (0.0) | 0 (0.0) | 0 (0.0) | 0 (0.0) | 0 (0.0) | 0 (0.0) | 0 (0.0) | 0 (0.0) | 1 (100.0) |
| | R | 0 (0.0) | 0 (0.0) | 1(100.0) | 0 (0.0) | 1 (100) | 0 (0.0) | 0 (0.0) | 0 (0.0) | 1 (100.0) | 0 (0.0) | 0 (0.0) |
| Total (14) | S | 12 (85.6) | 14 (100.0) | 6 (42.8) | 9(64.3) | 0 (0.0) | 13 (92.9) | 9 (64.3) | 14 (100.0) | 0 (0.0) | 8 (57.1) | 11 (78.5) |
| | I | 2 (14.4) | 0 (0.0) | 2 (14.4) | 1 (7.1) | 0 (0.0) | 1 (7.1) | 2 (14.3) | 0 (0.0) | 1 (7.1) | 2 (14.3) | 1 (7.1) |
| | R | 0 (0.0) | 0 (0.0) | 6 (42.8) | 4 (28.6) | 14 (100.0) | 0 (0.0) | 3 (21.4) | 0 (0.0) | 13 (92.9) | 4 (28.6) | 2 (14.4) |

MER = Meropenem, CIP = Ciprofloxacin, TOB = Tobramycin, AK = Amikacin, AMC = Amoxicillin/clavulanate, CN = Gentamycin, TE = Tetracycline, CAF = Chloramphenicol, AM = Ampicillin, DOX = Doxycycline, SXT = Trimethoprim/sulfamethoxazole, S = Sensitive, I = Intermediate, R = Resistant

**Table 7. Percentage of multiple drug resistance pattern of all bacterial isolates (n = 44) from Ayder Comprehensive Specialized Hospital from February to June 2019.**

| Bacterial isolates | Total n (%) | Antimicrobial resistance pattern | | | | | | |
|---|---|---|---|---|---|---|---|---|
| | | R0 | R1 | R2 | R3 | R4 | ≥ R5 | Multiple drug resistance |
| *Enterobacteriaceae* | 14 (31.8) | 0 (0.0) | 0 (0.0) | 5 (35.7) | 5 (35.7) | 1 (7.2) | 3 (21.4) | 8 (57.1) |
| *E. coli* | 11 (25.0) | 0 (0.0) | 0 (0.0) | 5 (45.4) | 4 (36.4) | 1 (9.1) | 1 (9.1) | 5 (45.5) |
| *Citrobacter* spp. | 2 (4.5) | 0 (0.0) | 0 (0.0) | 0 (0.0) | 0 (0.0) | 0 (0.0) | 2 (100.0) | 2 (100.0) |
| *K. pneumoniae* | 1 (2.3) | 0 (0.0) | 0 (0.0) | 0 (0.0) | 1(100.0) | 0 (0.0) | 0 (0.0) | 1 (100.0) |
| *Staphylococcus* strains | 30 (68.2) | 0 (0.0) | 7 (23.3) | 9 (30.0) | 4 (13.3) | 2 (6.7) | 8 (26.7) | 12 (40.0) |
| CoNS | 17 (38.6) | 0 (0.0) | 4 (23.5) | 5 (29.4) | 2 (11.8) | 2 (11.8) | 4 (23.5) | 8 (62.5) |
| *S. aureus* | 13 (29.6) | 0 (0.0) | 3 (23.0) | 4 (30.8) | 2 (15.4) | 0 (0.0) | 4 (30.8) | 4 (30.8) |
| Total | 44(100.0) | 0 (0.0) | 7 (15.9) | 14 (31.8) | 9 (20.5) | 3 (6.8) | 11 (25.0) | 20 (45.5) |

CoNS = Coagulase negative staphylococci, *S. aureus* = *Staphylococcus aureus*, *E. coli* = *Escherichia coli*, *K. pneumoniae* = *Klebsiella pneumonia*

R0: susceptible to all antibiotic, R1: resistant to 1 antibiotic, R2: resistant to 2 antibiotics, R3: resistant to 3 antibiotics, R4: resistant to 4 antibiotics, ≥ R5: resistant to 5 or more antibiotics

[35–38], Nigeria (17.3%) [39], South Africa (17.7%) [40], Kenya (19.4% and 23%) [41, 42], Tanzania (20.9%) [43], and Ethiopia (18.0% and 19.4%) [27, 29]. However, the current study showed higher prevalence of BV than several previous studies from different countries in which the incidence ranged from 2.3% in Uganda to 12.3% in the Burkina Faso [44–50]. In a meta-analysis by Torrone et al. [51], the estimated rate of BV among women in sub-Saharan Africa was 42.1% that was higher than our study. One of the reasons for this high prevalence could be due to the difference in the dominant microbial population living in the vagina of African women compared to other women. In a study by Fettweis et al. [52], African American women were found to be frequently colonized with *Gardnerella vaginalis* and the uncultivated bacterial vaginosis-associated bacterium-1 (BVAB1). These women are more prone to BV than women of European ancestry, who are more likely to harbor a *Lactobacillus*-dominated microbiome [52]. The prevalence of BV in this study was lower than previous reports from Nepal [35], Brazil [35, 53], Kenya [54, 55], Tanzania [56], Nigeria [23, 57, 58], Ghana [59], Cameroon [60], Algeria [61], Zimbabwe [62, 63], Egypt [64], Zambia [65], and Ethiopia [4], that ranged from 26.0%% to 60%. This may happen due to the difference in studied population or the test criteria for BV diagnosis [23, 35, 53–55, 57, 58, 64, 66]. Because, there are two main ways to diagnose BV: the Amsel clinical criteria and the microscopic Nugent criteria. Despite its high sensitivity and reproducibility, Nugent scoring is time-consuming, expensive, and requires lab equipment and specialists, which can cause great problems in developing countries. However, Amsel criteria are simple, fast, and inexpensive [66].

The cause of BV remains unclear and has been associated with demographic, sexual, reproductive health, and behavioral characteristics [67]. In the current study age, multiple sexual partners, HIV positivity, history of abortion and number of pantyliner used per day were not found significantly associated to BV. However, in contrast to the current findings, other studies showed that multiple lifetime sexual partner, women aged 45–64 years, women who used one pantyliner for two to four day, previous history of spontaneous abortion had significant association with prevalence of BV [4, 25, 27].

In this study, the prevalence of BV was higher among the women in second trimester than the women in first and third trimester of gestational age. There was a significant association between the BV and second trimester of gestational age in pregnant women [AOR (95% CI): 0.563 (0.324, 0.979), *P* = 0.042]. The current study disagreed with studies conducted in Ethiopia and Sudan that gestational age was not significantly associated with BV [8, 26, 27].

However, it was in agreement with the studies conducted elsewhere in Africa [39, 40, 59], in which an inverse association was seen towards the third trimester. In contrast to the current study, a study reported from South Africa, revealed that HIV-positive status was significantly associated with BV [40]. The present study showed that symptomatic pregnant women with white homogenous discharge had 2.7 times higher BV than asymptomatic pregnant women and BV was significantly associated with the symptoms of white homogenous discharge ($P < 0.001$) which was in line with a study conducted in Nigeria ($P = 0.001$) [39] but differed from a report from Ghana [59].

Another finding of the current study was the prevalence rate of 8.1% for AV in the pregnant women which was in agreement with the study conducted by Donders et al. [7] from Belgium (7.9%). The current prevalence was lower than studies reported from Ecuador 51.6% [50], Italy 60.4% [68], India 17.4% [32], 20.8% [69], 51.4% [70], China (15.4%) [71], and Bosnia and Herzegovina (51.0%) [72]. The disparity in the prevalence of AV in the current study with different studies may be due to the difference in test methodology, sample size, and the type of study population involved in the research. However, we did not found any study participant who had gonorrhea which was in good agreement with a study reported from Addis Ababa, Ethiopia by Bitew et al. [4] but differed from the study conducted in Bahir Dar, Ethiopia [8].

To the best of our knowledge, AV and its associated risk factors is not as extensively investigated as BV in pregnant women from Ethiopia and other countries and further research is needed to explore this connection. In this study, no socio-demographic, sexual, and behavioral characteristics variables were significantly associated with the prevalence of AV in pregnant women except for education and occupation status that was in contrast to previous study by Sianou et al. [73] from Greece. They found that AV was more frequent in prepubertal girls. In another study by Geng et al. [25] from China, unmarried status and frequent vaginal douching were among the high risk factors for AV in women, while the college-level education or above and regular condom use were both protective factors for AV. Perhaps one of the reasons for the high rate of AV in unmarried women is that they never or seldom use condoms during sexual activity [25, 74]. Also, frequent vaginal douching, especially during pregnancy, may be linked to AV by imbalance of the vaginal microflora or inflammation caused by physical or chemical irritation [25]. In a study by Hassan et al. [11] from Egypt, the education and occupation status were among the factors that were associated with the prevalence of AV in pregnant women that was in agreement with the current study. This may happen due to the difference in behavioral change towards risky sexual practices and knowledge in how to protect their personal hygiene. In contrast to the current study, Han et al. [75] found that a history of vaginal infection was a risk factor for AV during pregnancy. Consistent with the current study, Salinas et al. [50] and Pacha-Herrera et al. [76] from Ecuador, found no significant association between AV and any particular age group of the women studied.

Another finding of the current study, was the determination of bacterial isolates of AV and their antibiotic resistance. This research revealed the AV as a sole infection in 1.4% of pregnant women, while in 6.6% of cases, it was seen in mixed form with other infections. A previous study from China reported a higher prevalence rate of mixed AV cases with other infections [77]. In the current study, a total of 44 bacterial isolates were recovered, of which 30 (68.2%) and 14 (31.8%) isolates were staphylococci and *Enterobacteriaceae*, respectively. These findings were in contrast to a previous study conducted in India in which a greater predominance of *Enterobacteriaceae* isolates was reported [70]. Among all bacterial isolate, CoNS strains (38.6%) were the highest prevalent bacteria followed by *S. aureus* (29.5%) and *E. coli* (25.0%). Contrary to the current study, Sangeetha et al. [69] reported the *Enterococcus faecalis* (32.2%) as the most prevalent bacteria in AV women. In another report by Tang et al. [14] China, *E. coli* (32.4%) was the most frequent isolates in AV cases followed by *Staphylococcus* spp.

(21.8%) and *Enterococcus* spp. This study revealed the *E. coli* (78.6%) as the most predominant *Enterobacteriaceae* that was in line with previous studies [4, 14, 69, 70]. However, no *S. agalactiae* isolates were found in this study that was in contrast to previous studies in which the GBS strains had been reported with the frequency of 16.8% [14] and 9.6% [69]. One of the points to note is the inability of traditional techniques such as culture method to examine and detect all the bacteria involved in BV or AV cases, and many fastidious and slow-growing bacteria such as *Atopobium vaginae*, *Ureaplasma* and *Mycoplasma* species may not be identified and isolated in these infections using conventional methods. With the advent of high-throughput sequencing (next-generation sequencing) and molecular testing such as real-time polymerase chain reaction, the role of various pathogens including fastidious and slow-growing bacteria may become more apparent in BV and AV diseases [15, 50, 52, 76, 78, 79].

The overall resistance rates of *Enterobacteriaceae* were ranged from 0.0% for ciprofloxacin and chloramphenicol to 100.0% against amoxicillin/clavulanate. Also, gentamicin with a susceptibility rate of 92.9% was the third most effective antibiotic against *Enterobacteriaceae*. In line with this study, Sangeetha et al. [69], reported the ciprofloxacin and gentamicin among the most effective antibiotics against *Enterobacteriaceae*. The carbapenem category (imipenem) had also good efficacy against *Enterobacteriaceae* that was in parallel with the previous reports from China and India [14, 70]. The overall resistance rate of Gram-positive bacteria was observed against penicillin (100.0%), followed by erythromycin (60.0%). However, 86.7% of the isolates were sensitive to ciprofloxacin followed by gentamicin (83.3%). These results confirmed that ciprofloxacin can be used in treatment of AV in our region. As a result of their little effect on the normal microflora, fluoroquinolones such as ciprofloxacin are commonly used to treat AV because they allow rapid recovery from the condition [15]. However, the use of some antibiotics such as quinolones has been contraindicated during pregnancy due to concerns of carcinogenesis and fetal malformations in animals and there is conflicting evidence regarding their safety in humans [80]. *S. aureus*, the 2nd most frequently isolated Gram-positive bacterium was 92.3% susceptible to ciprofloxacin followed by chloramphenicol (84.6%) and gentamicin (76.9%). However, *S. aureus* had a high resistance rate for penicillin (100.0%), erythromycin (53.8%) and trimethoprim/sulfamethoxazole (53.8%). Our result was consistent with other studies conducted in Addis Ababa and Bahir Dar [4, 8]. Today, there is no consensus among clinicians regarding how to treat AV. Antibiotics that are intrinsically active against fecal bacteria and also have minimal effects on vaginal *Lactobacillus* species are the best choices for the treatment of AV. However, the amount of inflammation typically associated with AV may make antibiotics ineffective for most patients with this condition [15, 81].

In the current study, the frequency of multiple drug resistant isolates was higher (45.5%) than previous study by Tang et al. [14] from China (13.4%). In recent years, the prevalence of infectious diseases and multiple drug resistant bacteria harboring different resistance genes has been increasing [82–85]. These issues highlight the need to develop a global monitoring program, especially for pregnant women, to reduce the risk of vaginal infection spreading among the mothers and infants and to control the antibiotic resistance phenomenon in different countries.

This study had several limitations. The lack of nonpregnant control women in this study made it impossible to accurately determine the risk factors associated with BV and AV in pregnant women. The lack of culture results from a comparison group of healthy pregnant women raises the question of whether bacterial isolates from AV are truly pathogens or just colonizers. The isolation of bacterial species by conventional culture method and the lack of molecular methods result in the loss of some bacterial species. The descriptive cross-sectional nature of the study rather than a cohort lead to no subsequent assessment of the 422 women. Finally, the

lack of some information such as ethnicity of the population studied, made it impossible to make a comparison in this regard.

## Conclusion

In summary, the overall prevalence of BV and AV among pregnant women was 20.1% and 8.1%, respectively. The prevalence of BV was higher among symptomatic (35.1%) than asymptomatic pregnant women (16.8%). White vaginal discharge and second trimester were the risk factor and protective item for BV, respectively. Whereas, secondary school and housewife were also found protective and risk factor for AV, respectively. The *Staphylococcus* spp. were more prevalent than the *Enterobacteriaceae* in AV pregnant women. Amoxicillin/clavulanate had the highest overall resistance rate against *Enterobacteriaceae*, while the overall resistance rate of *Staphylococcus* spp. was observed against penicillin. In view of the prevalence of vaginal infections among pregnant women, prompt and adequate investigations with appropriate treatment are needed to prevent the adverse effect of the infection on mother and fetus. They should pay additional attention to education on sexual hygiene measures and sexual risk behaviors.

## Supporting information

**S1 File. Research plan questionnaire.**
(PDF)

**S1 Table. Socio-demographic characteristics of pregnant women at Ayder comprehensive Specialized Hospital from February to June 2019.**
(DOCX)

**S2 Table. Prevalence of vaginal infections among pregnant women at Ayder Comprehensive Specialized Hospital from February to June 2019.**
(DOCX)

**S3 Table. Univariate and multivariate analysis of factors associated with bacterial vaginosis among pregnant women attending antenatal care in Ayder Comprehensive Specialized Hospital from February to June 2019.**
(DOCX)

**S4 Table. Univariate and multivariate analysis of factors associated with aerobic vaginitis among pregnant women attending antenatal care in Ayder Comprehensive Specialized Hospital from February to June 2019.**
(DOCX)

**S5 Table. Percentage of antibacterial susceptibility pattern of all *Staphylococcus* isolates (n = 30) from Ayder Comprehensive Specialized Hospital from February to June 2019.**
(DOCX)

**S6 Table. Percentage of antibacterial susceptibility pattern of all *Enterobacteriaceae* isolates (n = 14) from the Ayder Comprehensive Specialized Hospital from February to June 2019.**
(DOCX)

**S7 Table. Percentage of multiple drug resistance pattern of all bacterial isolates (n = 44) from Ayder Comprehensive Specialized Hospital from February to June 2019.**
(DOCX)

## Acknowledgments

We acknowledge Mekelle University College of Health Science, Biomedical laboratory and Ayder Comprehensive Specialized Hospital Microbiology Unit for allowing the use of laboratory equipment and technical support.

## Author Contributions

**Conceptualization:** Gebrehiwet Tesfay Yalew, Saravanan Muthupandian, Gopinath Venkatraman, Yemane Mengsteab Hagos, Hadush Negash Meles, Hagos Haileslasie Weldehaweriat, Morteza Saki.

**Data curation:** Gopinath Venkatraman.

**Formal analysis:** Gebrehiwet Tesfay Yalew, Saravanan Muthupandian, Kiflom Hagos, Gopinath Venkatraman, Morteza Saki.

**Investigation:** Gebrehiwet Tesfay Yalew, Saravanan Muthupandian, Letemichael Negash, Yemane Mengsteab Hagos, Hussein O. M. Al-Dahmoshi.

**Methodology:** Gebrehiwet Tesfay Yalew, Saravanan Muthupandian, Kiflom Hagos, Letemichael Negash, Gopinath Venkatraman, Yemane Mengsteab Hagos, Hadush Negash Meles, Hagos Haileslasie Weldehaweriat, Hussein O. M. Al-Dahmoshi.

**Resources:** Kiflom Hagos.

**Software:** Hadush Negash Meles, Hagos Haileslasie Weldehaweriat, Morteza Saki.

**Supervision:** Gebrehiwet Tesfay Yalew, Saravanan Muthupandian, Kiflom Hagos, Letemichael Negash, Gopinath Venkatraman, Yemane Mengsteab Hagos, Hagos Haileslasie Weldehaweriat.

**Validation:** Gebrehiwet Tesfay Yalew, Kiflom Hagos, Letemichael Negash, Hadush Negash Meles, Hagos Haileslasie Weldehaweriat.

**Visualization:** Letemichael Negash, Yemane Mengsteab Hagos, Hadush Negash Meles, Hussein O. M. Al-Dahmoshi.

**Writing – original draft:** Gebrehiwet Tesfay Yalew, Saravanan Muthupandian, Kiflom Hagos, Letemichael Negash, Gopinath Venkatraman, Yemane Mengsteab Hagos, Hagos Haileslasie Weldehaweriat, Hussein O. M. Al-Dahmoshi, Morteza Saki.

**Writing – review & editing:** Gebrehiwet Tesfay Yalew, Saravanan Muthupandian, Letemichael Negash, Gopinath Venkatraman, Hussein O. M. Al-Dahmoshi, Morteza Saki.

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
