## [Decision Letter · Decision Letter 0]

19 Oct 2021

PONE-D-21-28305Prevalence of bacterial vaginosis and aerobic vaginitis and their associated risk factors among pregnant women from northern Ethiopia: a cross-sectional studyPLOS ONE

Dear Dr. Saki,

Thank you for submitting your manuscript to PLOS ONE. After careful consideration, we feel that it has merit but does not fully meet PLOS ONE’s publication criteria as it currently stands. Therefore, we invite you to submit a revised version of the manuscript that addresses the points raised during the review process. 1. Please state more clearly the underlying hypothesis and pre-specified analyses, as this will frame the results a bit better.2. Please try and streamline the results, and consider consolidating or eliminating tables3. Please consider seriously the reviewer comments about the bacterial culture data. This section of the paper seems unrelated to the title, and has some significant limitations in terms of what it can tell readers about aerobic vaginitis.

We look forward to receiving your revised manuscript.

Kind regards,

Caroline Mitchell

Academic Editor

PLOS ONE

Journal Requirements:

“This research was financed by Mekelle University, Mekelle -1871, Ethiopia.”

Additional Editor Comments:

In this paper the authors describe the prevalence of bacterial vaginosis, diagnosed by gram stain and aerobic vaginitis, diagnosed by wet mount in a cohort of 422 pregnant women in Ethiopia. Analyses were performed to evaluate which demographic characteristics were associated with each diagnosis. Additionally, the authors cultivated bacteria from women with AV and tested antimicrobial resistance patterns. It is unclear to me what this second set of data add to the primary focus of the manuscript as outlined in the title, which references prevalence and risk factors.

In the introduction the authors seem to equate the association between BV and poor pregnancy outcomes and that of AV and poor pregnancy outcomes. There are FAR less data to suggest that AV is associated with adverse pregnancy outcomes, and I would recommend changing the introduction to reflect this. (For example, in the review cited as reference 12, the authors did not identify ANY studies of AV and adverse outcomes in SSA).

On line 76, the authors state that: “having a history of spontaneous abortion and altered vaginal bacterial communities may contribute to vaginal colonization with microorganisms that are related to endogenous infection such as BV or AV”. The altered vaginal bacterial community is BV, and so people with this are already colonized with the BV-associated organisms, rather than the link from one to the other implied in the sentence. None of the references for this statement describe an association between history of SAB and BV or AV, nor am I aware of such data in the literature. BV may increase the risk for SAB, but this has not been a consistent association.

On line 153-156, it would be ideal to specify that the scale goes the opposite direction for Lactobacillus morphotypes – i.e. that no morphotypes = 4)

How was the sample size arrived at? Is this a convenience sample? Was a power calculation done prior to the study?

How were variables for the multivariable analyses selected? A priori? (For BV, marital status is included, but was not significant in univariate analysis).

What was the underlying hypothesis of the study? Did the authors undertake the study because they thought that BV and AV might have different prevalence in Ethiopia? Or different risk factors? Stating this up front would help frame the study and the discussion of the results – then the statements about how the results are similar to or different from prior studies are linked to a hypothesis and either support that hypothesis or don’t.

In the section from 201-208, it is a bit confusing to state the total with normal scores for both BV and AV, and then to restate the individual BV and AV scores, because the numbers overlap a bit. The individual scores are outlined in Table 1, so it might be fine to cite Table 1 after the first sentence of that paragraph and then to say something like: YYY had normal scores for both BV and AV, YYY had both AV and BV, and YYY had one or the other.(Table 2). Ideally you don’t need to repeat information in the text that is present in a table.

The results spend a lot of time discussing results that are not statistically significant. The discussion implies that the authors think they really are meaningful, but perhaps that the study was underpowered to detect a difference? Since all of the information is in Table 3, much of this discussion could be cut, and only those variables discussed that the authors think are truly biologically relevant, but perhaps the analysis is underpowered.

On line 283-284 the authors say that bacteria were isolated from women with “polymicrobial infection.” The title of the section is about isolation from women with AV – so I think I would stick with that terminology. Adding the term/concept polymicrobial infection here is problematic – in part because it isn’t clear that AV is really an infection, and if we consider the type of community disruption seen with AV a polymicrobial infection, then BV would be that as well.

Were cultures done for women without AV? How do the authors know that the isolates found in women with AV are causal? What is the prevalence in women without AV?

Lines 312-321 include several sentences that seem to be only partial ideas, or that are incomplete. Please revise to make the narrative flow more smoothly.

Between lines 312-335 the authors spend a great deal of time discussing how the prevalence of BV in this study is similar or different to other studies and populations. This could be condensed significantly.

Lines 336-364 also spend a great deal of time discussing which risk factors were and were not associated with BV, and this discussion could be condensed. These two sections, and the section that follow these, about risk factors for AV, would be better if focused more tightly on what seem to be the key questions: Are risk factors for BV and AV the same? Is there anything dramatically different about those risk factors in pregnant Ethiopian women than other populations?

The section in the discussion about the various bacterial isolates does not discuss some important considerations: 1) the many bacteria that may be present but not easily cultivable, as has been shown for BV and 2) whether AV even needs to be treated. There is an extensive discussion about antibiotics, and use of fluoroquinolones – but these are not acceptable for use in pregnancy. As noted in my comments above, the lack of culture results from a comparison group of healthy pregnant women begs the question of whether these isolates are truly pathogens, or just colonizers.

In the conclusion, the authors recommend screening pregnant women for AV and BV, however, no studies have shown benefit to screening and treatment of asymptomatic BV in pregnancy. What is the rationale for recommending screening?

In the text, and in table 1 – there is a variable that is “Number of pants used per day” – the meaning of this isn’t clear to the general reader. Does it mean undergarment? Pantyliner? Something else?

In Table 2, it would be ideal to specify in a footnote how each diagnosis was determined, so that the table can stand alone: i.e. BV = Nugent 7-10, AV = score > 2, Yeast = + wet mount, etc.

In Tables 1, 3 and 5 the variable “contraceptive use” should be specified – does this mean hormonal contraceptive use? Ever use? All of the participants are pregnant, so when is the period of contraceptive use that was queried?

I’m not sure I think table 4 and table 6 are necessary – it doesn’t seem to contribute to the overall narrative. However, defer to the authors here – I don’t feel strongly about this.

Reviewers' comments:

Reviewer's Responses to Questions

**Comments to the Author**

1. Is the manuscript technically sound, and do the data support the conclusions?

Reviewer #1: Partly

2. Has the statistical analysis been performed appropriately and rigorously? 

Reviewer #1: Yes

3. Have the authors made all data underlying the findings in their manuscript fully available?

Reviewer #1: Yes

4. Is the manuscript presented in an intelligible fashion and written in standard English?

Reviewer #1: No

5. Review Comments to the Author

Reviewer #1: Congratulations to the authors for the present manuscript. It is a relevant study in vaginal microbiota.

However, major revisions are needed before publication due to the excesive number of tables (sometimes unnecessary), bold statements, lack of the some discussion about some results and comparisons with others study about BV and AV, and finally there are no recognition of the limitations of the study by the authors referring only the strengthens of the study.

Therefore, major revisions are needed in order to approve the present manuscript for publication. Please, I kindly invite the authors to review the following comments:

GENERAL COMMENTS

1. The AV classification and biochemical identificacion of the bacteria species are not cleary specified in Material and Methods of the manuscript.

2. Results showed an excess of the tables that could be easily simplify to the Readers, therefore some adjustments are needed by the authors.

3. Some recent studies on Pregnant Women and AV are lacking in the discussion of the manuscript, such as:

https://www.frontiersin.org/articles/10.3389/fcimb.2020.00303/full

https://www.nature.com/articles/s41598-020-74655-z

So, I recommend the authors to udpate the discussion with at least the examples given above.

4. Bold statements are made by authors in several parts of the study, such as bacteria profiles (the isolation of the bacteria was made with selective growth media, lacking the possibility to isolate other bacterial species and so it is not bacterial profile), the MDR of the bacterial isolates (misleading the Readers) and referring Staphylococcus especies as GPB and genera of Enterobacteriaceae as GNB (which is an incorrect generalization of the results).

5. Several strengthens are appointed by the authors during the manuscript, but no limitations are given by them. There are many limitations in the study that need to be recognized. Some examples are:

- Lack of non-pregnant women in the study unable to detect strictly risky factors associated with pregnant women with BV or AV.

- The isolation of bacteria species were conducted by selective media and therefore part of the bacteria profiles could be easily lost in the process.

- It is a cross-sectional study and no following assessment was made to the 422 women.

- No information about the ethnicity of the population set was given to the Readers.

6. Overall, the manuscript showed a lack of English editing, showing several grammatical and orthographic errors.

ABSTRACT

Line 35- Please add "bacterial" before vaginosis.

Line 48- Please writte the full name of E. coli because it is the first time that you cite the species (just as you did before with Staphylococcus aureus).

Line 51- If you decide to put 1 decimal in the previous percentage numbers, you should mantain it in all values of the manuscript.

INTRODUCTION

Lines 66-67- Both AV and BV are vaginal dysbiosis and so they are characterized by the lactobacilli reduction. Please, rephrase the sentence.

Line 74- Please add the reference of Pacha-Herrera et al. (2020) about BV and AV in pregnant and non-pregnant women. Front. Cell. Infect. Microbiol., 19 June 2020 | https://doi.org/10.3389/fcimb.2020.00303

Line 78- Please replace the term infection by dysbiosis.

Line 83- Please state the full name of HIV before write its abbreviation, as you did with HSV-2.

Line 101- Please remove the bacterial etiology as a goal of the study, it is a bold statement. A more appropriate statment is a vaginal microbiota evaluation of the vaginal samples.

General comment of this section- You should edit the English text.

MATERIALS AND METHODS

Line 113- Please add the abbreviation "ACSH" after the full name of "Ayder Comprehensive Specialized Hospital".

Line 118- Please add the "" ext-link-type="uri" xlink:type="simple">http://www.mu.edu.et/chs/index.php/ayder-refferal-hospital" as reference. Also, I could not access the online page link. Is it correct?

Line 129- It is a limitation to only exclude women with antibiotic therapy in the last two weeks, because it takes more than two weeks to recover the normal vaginal microbiota. It should be add as a limitation of the study.

Line 131- The women under legal age were also excluded of the study, correct?

Line 140- The authors cite the Table 2 before Table 1. Thefore, I recommend to delete this reference and allow the correct sequence of citations of the Tables in Results section.

Line 145- Please mantain one type of magnification, it seems that sometimes the authors referred the objective 40x and other times cited the total magnification (as in the line 161 with 400x magnification).

Line 148- Please rephrase the first sentence, double stained in the same sentence can be avoid.

Line 151- Please add spp. after Lactobacillus.

Line 151- Please replace "Gardnerella vaginalis" by "Gardnerella/Bacteroides spp."

Line 152- Please delete "s" in "spps.".

Line 152- Why Gram-positive cocci? Please reviewe Nugent score: https://www.ncbi.nlm.nih.gov/pmc/articles/PMC269757/

Line 156- Why intermediate BV in Nugent score between 4 and 6? It is a intermediate vaginal microbiota that did not achieve a vaginal dysbiosis yet. It is incorrect to cite as intermediate BV.

Lines 162-163- Please a further explaination of the AV score is needed for the Readers to clarify the Av diagnosis.

Line 164- Please change the title because it is incorrect. The section described the isolation of certain bacteria species from vaginal samples.

Lines 171-172- Please change 2 in underscript in "H2S" and to add the reference of the classification of the bacterial isolates through the biochemical evaluation.

Line 173- Please avoid the abbreviation in the titles. I recommend to do it in the first sentence of the section (line 174).

Line 186- Please replace "Collected quantitative data were coded; processed, edited, and analyzed" by "Collected quantitative data were coded and analyzed".

Line 189- Please replace "(IC)" by "(CI)".

RESULTS

Line 195- Please replace "for BV and AV" by "for their vaginal microbiota".

Line 203- Please replace "yeast cell infection" by "candidiasis" and "Trichomonas vaginalis" by "trichomoniasis (Trichomonas vaginalis)".

Lines 202-210- Please avoid the repetition of "422 study participants" and mantain the results form of presentation. I recommend to cite the percentagen (number of cases positive/total number of cases).

Lines 213-215- Please rephrase the sentence. The prevalences of BV between age categories are similar (21.6, 20.7 and 18.3%) and without statistical differences.

Lines 228-230- Please rephrase the sentence. The prevalences of BV are similar and without statistical differences.

Line 241- Please add "0" in "P = .042".

Lines 244-249- Please explain better the results and rectify "P = .042".

Lines 250-251- Please explain better. I am a little confused... the prevalence of BV in symptomatic pregnant women was 31.8% and the prevalence of BV in asymptomatic women was 68.2% in Table 4.

Lines 260-265- Please rephrase the sentences.

Lines 273-275- Please rephrase the sentence and rectify "[0.292 (0.102, 0.833)], (P = 0.021))".

Line 274- What do you mean with "may significantly associate"?

Lines 279-281- Please rephrase the sentence.

Line 283- Please clarify to the readers that 44 bacterial isolates are from AV women. BV and AV are both vaginal dysbiosis.

Lines 285 until the remaining manuscript- The abreviations GPB (Gram PositiveBacteria) and GNB (Gram Negative Bacteria) were added directly without the full names, but most important they are overgeneralize terms misleading the Readers. All GPB are Staphylococcus genus and all GNP are genera of Enterobacteriaceae family. Please rectify these terms in all manuscript.

Lines 288, 293, and 299- I recommend to delete these titles because you did not possess so many results and these 3 paragraphs can be add after the only paragrah after the title "Bacterial isolates among pregnant women with aerobic vaginitis".

Lines 302-304- "... including Citrobacter spp. and K. pneumoniae were 100.0% multidrug-resistant (MDR)". This sentence are misleading the Readers. There are only 2 isolates of Citrobacter spp. and just one K. pneumoniae showing also sensitive pattern to antibiotics. Please clarify better this information to the Readers and I recommend to do it also in Staphylococcus isolates.

DISCUSSION

Line 309 and the remaining discussion- Please add also another study with BV and AV in Latin America women, more exactly: Ecuador - Salinas et al (2020) https://www.nature.com/articles/s41598-020-74655-z

Lines 312-314- Please put each individual reference to each country.

Lines 333-335- I would invite the authors to further discuss the discrepancies between studies, besides the diagnostic methods. It seems too vague.

Lines 348-350- I would invite the authors to further discuss the statistical significance of the results. It is too vague.

Lines 356-358- Why it is relevant to the discussion? It seems too vague.

Lines 368-370- I would invite the authors to further discuss the discrepancies between studies. It seems too vague.

Lines 382- I would invite the authors to also compare the prevalence of AV in age with Salinas et al (2020) and Pacha-Herrera et al. (2020):

https://www.nature.com/articles/s41598-020-74655-z

https://doi.org/10.3389/fcimb.2020.00303

Lines 397-398- I would invite the authors to further discuss the differences in lifestyle between these population subsets. It seems too vague.

Lines 398-400- Please compare the results of Han et al. (2019) with your results.

Lines 401-436- Please rewrite this subsection of discussion. There no bacterial profile done in present study, GPB and GNB terms are misleading the Readers.

Lines 413-415- Please state the prevalence of S. agalactiae of these studies.

CONCLUSION

Lines 439-440- I am a little confused. In table 4, the prevalence of BV was higher in asymptomatic women (68.2%) than in symptomatic women (31.8%) and contradicting the lines 449 and 450 of the Conclusion.

I suggest the authors to rewrite the Conclusion section and to add the limitations of the study, which there are several as previously referred in the GENERAL COMMENTS.

TABLES

There are too many tables in the manuscript (10 tables) and the authors can easily reduce the number of tables.

Tables 1 and 2 can easily become one table. I recommend the authors to see the Table 1 of Salinas et al. (2020) and adapt their Tables 1 and 2 in that format (https://www.nature.com/articles/s41598-020-74655-z/tables/1).

Table 2 did not show intermediate vaginal microbiota.

Table 4 showed "no" instead of "n", as previously referred in other tables.

Table 6 showed "aerobic vaginosis" instead of "aerobic vaginitis".

Table 7 showed information that is repeated in table 10.

Tables 8, 9 and 10 could be combined in one unique table.

General remarks of the tables: * = Significant association can be improve by indicating the statistical analysis made (such as Table 4). Also, the page numbers are wrong after page 30 of the manuscript.

6. PLOS authors have the option to publish the peer review history of their article (what does this mean?). If published, this will include your full peer review and any attached files.

Reviewer #1: **Yes: **José António Baptista Machado Soares

---

## [Author Response · Author response to Decision Letter 0]

11 Nov 2021

Dear Dr. Caroline Mitchell 

Academic Editor of PLOS ONE

Subject : Submission of revised manuscript entitled "Prevalence of bacterial vaginosis and aerobic vaginitis and their associated risk factors among pregnant women from northern Ethiopia: a cross-sectional study" (ID: PONE-D-21-28305).

Thank you for your email of 19 October, 2021 enclosing the editor and reviewers ʼcomments.We also greatly appreciate the reviewers for their complimentary comments and suggestions. We have carefully reviewed the comments and have revised the manuscript accordingly. Our point to point responses are given below. (The reviewer’s comments are in italics). Changes to the manuscript are indicated in yellow highlighted sentences/words. We hope that you find our responses satisfactory and that the manuscript is now acceptable for publication. Anyway, we should be grateful if you let us know about our further changes required.

Yours sincerely,

Corresponding author: Morteza Saki

Mailing address: Department of Microbiology, Faculty of Medicine, Ahvaz Jundishapur University of Medical Sciences, Ahvaz, Iran

P.O. Box: 159 

Postal Code: 61357-15794 

Email: mortezasaki1981@gmail.com

The followings are point-by-point responses :

Journal Requirements:

Response (R): We revised the manuscript and confirm that the revised manuscript is according to the journal style format and requirements. Also, we placed all tables in appropriate places.

“This research was financed by Mekelle University, Mekelle -1871, Ethiopia.”

R: We removed the “This research was financed by the Mekelle University, Mekelle -1871, Ethiopia.” From Acknowledgments. However, we would like to remain this sentences in the funding section of the published manuscript, if it get accepted. Also, we add this in cover letter.

R: We add “All relevant data are within the paper.” in the submission system.

Additional Editor Comments:

In this paper the authors describe the prevalence of bacterial vaginosis, diagnosed by gram stain and aerobic vaginitis, diagnosed by wet mount in a cohort of 422 pregnant women in Ethiopia. Analyses were performed to evaluate which demographic characteristics were associated with each diagnosis. Additionally, the authors cultivated bacteria from women with AV and tested antimicrobial resistance patterns. It is unclear to me what this second set of data add to the primary focus of the manuscript as outlined in the title, which references prevalence and risk factors.

R: Since, the available data about the prevalence of bacterial pathogens involved in AV is scarce in many countries including Ethiopia, we performed the culture and antibiotic susceptibility tests for bacteria that collected from AV patients. Since the title may be too long, we could not reflect this issue in the title of manuscript. To the best of our knowledge, this study was the first to report the bacterial pathogens involved in AV cases from Ethiopia.

 In the introduction the authors seem to equate the association between BV and poor pregnancy outcomes and that of AV and poor pregnancy outcomes. There are FAR less data to suggest that AV is associated with adverse pregnancy outcomes, and I would recommend changing the introduction to reflect this. (For example, in the review cited as reference 12, the authors did not identify ANY studies of AV and adverse outcomes in SSA).

R: As we mentioned in the introduction, there is little data on the exact role of AV and poor pregnancy outcomes and this need further studies. We add some aspects regarding the cause of association of AV and poor pregnancy outcomes in the introduction. Also, we add a more suitable reference.

On line 76, the authors state that: “having a history of spontaneous abortion and altered vaginal bacterial communities may contribute to vaginal colonization with microorganisms that are related to endogenous infection such as BV or AV”. The altered vaginal bacterial community is BV, and so people with this are already colonized with the BV-associated organisms, rather than the link from one to the other implied in the sentence. None of the references for this statement describe an association between history of SAB and BV or AV, nor am I aware of such data in the literature. BV may increase the risk for SAB, but this has not been a consistent association.

R: We rephrase the sentences and corrected them accordingly. These sentences including “association between history of SAB and BV or AV “ are mentioned in reference 15 page 2. 

On line 153-156, it would be ideal to specify that the scale goes the opposite direction for Lactobacillus morphotypes – i.e. that no morphotypes = 4)

R: We add this in the materials and methods (page 16, line 159).

How was the sample size arrived at? Is this a convenience sample? Was a power calculation done prior to the study?

R: The sample size of this study was determined using the convenience sampling as a pilot design.

How were variables for the multivariable analyses selected? A priori? (For BV, marital status is included, but was not significant in univariate analysis).

R: An a priori selected set of variables with a P-value 0.2 in the univariate analysis were considered for the multivariate regression analysis.

What was the underlying hypothesis of the study? Did the authors undertake the study because they thought that BV and AV might have different prevalence in Ethiopia? Or different risk factors? Stating this up front would help frame the study and the discussion of the results – then the statements about how the results are similar to or different from prior studies are linked to a hypothesis and either support that hypothesis or don’t.

R: The current study was an epidemiologic cross-sectional study to shed more light on the prevalence of AV and BV and their associated risk factors from Northern Ethiopia. Because there are scarce data in this regard. This study was not an hypothesis to approve or disapprove it.

In the section from 201-208, it is a bit confusing to state the total with normal scores for both BV and AV, and then to restate the individual BV and AV scores, because the numbers overlap a bit. The individual scores are outlined in Table 1, so it might be fine to cite Table 1 after the first sentence of that paragraph and then to say something like: YYY had normal scores for both BV and AV, YYY had both AV and BV, and YYY had one or the other.(Table 2). Ideally you don’t need to repeat information in the text that is present in a table.

R: We made the correction accordingly and transferred the Tables in appropriate place.

The results spend a lot of time discussing results that are not statistically significant. The discussion implies that the authors think they really are meaningful, but perhaps that the study was underpowered to detect a difference? Since all of the information is in Table 3, much of this discussion could be cut, and only those variables discussed that the authors think are truly biologically relevant, but perhaps the analysis is underpowered.

R: We shorten the results accordingly.

On line 283-284 the authors say that bacteria were isolated from women with “polymicrobial infection.” The title of the section is about isolation from women with AV – so I think I would stick with that terminology. Adding the term/concept polymicrobial infection here is problematic – in part because it isn’t clear that AV is really an infection, and if we consider the type of community disruption seen with AV a polymicrobial infection, then BV would be that as well.

R: We corrected the term accordingly.

Were cultures done for women without AV? How do the authors know that the isolates found in women with AV are causal? What is the prevalence in women without AV?

R: We did not perform the bacterial culture for women without AV. We just determine the prevalence of bacterial pathogens isolated from AV cases and their antibiotic resistance patterns like several previous studies (references 69,71). 

Lines 312-321 include several sentences that seem to be only partial ideas, or that are incomplete. Please revise to make the narrative flow more smoothly.

R: We revised the lines 312 to 321 accordingly.

Between lines 312-335 the authors spend a great deal of time discussing how the prevalence of BV in this study is similar or different to other studies and populations. This could be condensed significantly.

R: We revised the lines 312 to 335 and condensed them accordingly.

Lines 336-364 also spend a great deal of time discussing which risk factors were and were not associated with BV, and this discussion could be condensed. These two sections, and the section that follow these, about risk factors for AV, would be better if focused more tightly on what seem to be the key questions: Are risk factors for BV and AV the same? Is there anything dramatically different about those risk factors in pregnant Ethiopian women than other populations?

R: We revised the lines 336 to 364 and condensed them accordingly.

The section in the discussion about the various bacterial isolates does not discuss some important considerations: 1) the many bacteria that may be present but not easily cultivable, as has been shown for BV and 2) whether AV even needs to be treated. There is an extensive discussion about antibiotics, and use of fluoroquinolones – but these are not acceptable for use in pregnancy. As noted in my comments above, the lack of culture results from a comparison group of healthy pregnant women begs the question of whether these isolates are truly pathogens, or just colonizers.

R: We add the some discussion about the many bacteria that may be present but not easily cultivable and wether AV need to be treated. Also, we add the “the lack of culture results from a comparison group of healthy pregnant women begs the question of whether these isolates are truly pathogens, or just colonizers” as a limitation of the current study.

In the conclusion, the authors recommend screening pregnant women for AV and BV, however, no studies have shown benefit to screening and treatment of asymptomatic BV in pregnancy. What is the rationale for recommending screening?

R: We deleted the claimed sentences.

In the text, and in table 1 – there is a variable that is “Number of pants used per day” – the meaning of this isn’t clear to the general reader. Does it mean undergarment? Pantyliner? Something else?

R: We corrected all pants accordingly and changed them to pantyliner.

In Table 2, it would be ideal to specify in a footnote how each diagnosis was determined, so that the table can stand alone: i.e. BV = Nugent 7-10, AV = score 2, Yeast = + wet mount, etc.

R: We added the scores accordingly.

In Tables 1, 3 and 5 the variable “contraceptive use” should be specified – does this mean hormonal contraceptive use? Ever use? All of the participants are pregnant, so when is the period of contraceptive use that was queried?

R: We deleted this factor from all manuscript.

I’m not sure I think table 4 and table 6 are necessary – it doesn’t seem to contribute to the overall narrative. However, defer to the authors here – I don’t feel strongly about this.

R: We deleted tha tables 4 and 6 accourdingly.

Reviewers' comments:

GENERAL COMMENTS

1. The AV classification and biochemical identificacion of the bacteria species are not cleary specified in Material and Methods of the manuscript.

R: As you know each test including AV score interpretation and several different biochemical tests are described in full detail in several studies previously published. Preferably, we used suitable references here to avoid lengthening the text. All readers can refer to the mentioned references for more information.

2. Results showed an excess of the tables that could be easily simplify to the Readers, therefore some adjustments are needed by the authors.

R: We deleted some tables accordingly.

3. Some recent studies on Pregnant Women and AV are lacking in the discussion of the manuscript, such as:

https://www.frontiersin.org/articles/10.3389/fcimb.2020.00303/full

https://www.nature.com/articles/s41598-020-74655-z

So, I recommend the authors to udpate the discussion with at least the examples given above.

R: We added the mentioned studies in the discussion accordingly.

4. Bold statements are made by authors in several parts of the study, such as bacteria profiles (the isolation of the bacteria was made with selective growth media, lacking the possibility to isolate other bacterial species and so it is not bacterial profile), the MDR of the bacterial isolates (misleading the Readers) and referring Staphylococcus especies as GPB and genera of Enterobacteriaceae as GNB (which is an incorrect generalization of the results).

R: We revised and corrected the mentioned statements.

5. Several strengthens are appointed by the authors during the manuscript, but no limitations are given by them. There are many limitations in the study that need to be recognized. Some examples are:

- Lack of non-pregnant women in the study unable to detect strictly risky factors associated with pregnant women with BV or AV.

- The isolation of bacteria species were conducted by selective media and therefore part of the bacteria profiles could be easily lost in the process.

- It is a cross-sectional study and no following assessment was made to the 422 women.

- No information about the ethnicity of the population set was given to the Readers.

R: We add all mentioned limitations in the conclusion.

6. Overall, the manuscript showed a lack of English editing, showing several grammatical and orthographic errors.

R: We revised the manuscript again and remove the grammatical errors. We try to send the best version of manuscript with out grammatical errors. 

ABSTRACT

Line 35- Please add "bacterial" before vaginosis.

R: Fixed.

Line 48- Please writte the full name of E. coli because it is the first time that you cite the species (just as you did before with Staphylococcus aureus).

R: Fixed.

Line 51- If you decide to put 1 decimal in the previous percentage numbers, you should mantain it in all values of the manuscript.

R: Fixed.

INTRODUCTION

Lines 66-67- Both AV and BV are vaginal dysbiosis and so they are characterized by the lactobacilli reduction. Please, rephrase the sentence.

R: Fixed.

Line 74- Please add the reference of Pacha-Herrera et al. (2020) about BV and AV in pregnant and non-pregnant women. Front. Cell. Infect. Microbiol., 19 June 2020 | https://doi.org/10.3389/fcimb.2020.00303

R: We used this reference in the discussion.

Line 78- Please replace the term infection by dysbiosis.

R: We rephrase all sentences according to editor comments.

Line 83- Please state the full name of HIV before write its abbreviation, as you did with HSV-2.

R: Fixed.

Line 101- Please remove the bacterial etiology as a goal of the study, it is a bold statement. A more appropriate statment is a vaginal microbiota evaluation of the vaginal samples.

R: Fixed.

General comment of this section- You should edit the English text.

R: We revised the manuscript again and remove the grammatical errors. We try to send the best version of manuscript with out grammatical errors. 

MATERIALS AND METHODS

Line 113- Please add the abbreviation "ACSH" after the full name of "Ayder Comprehensive Specialized Hospital".

Line 118- Please add the "http://www.mu.edu.et/chs/index.php/ayder-refferal-hospital" as reference. Also, I could not access the online page link. Is it correct?

R: We deleted the link.

Line 129- It is a limitation to only exclude women with antibiotic therapy in the last two weeks, because it takes more than two weeks to recover the normal vaginal microbiota. It should be add as a limitation of the study.

R: We add this as limitation.

Line 131- The women under legal age were also excluded of the study, correct?

R: We add this in exclusion criteria.

Line 140- The authors cite the Table 2 before Table 1. Thefore, I recommend to delete this reference and allow the correct sequence of citations of the Tables in Results section.

R: We corrected it accordingly.

Line 145- Please mantain one type of magnification, it seems that sometimes the authors referred the objective 40x and other times cited the total magnification (as in the line 161 with 400x magnification).

R: Fixed.

Line 148- Please rephrase the first sentence, double stained in the same sentence can be avoid.

R: Fixed.

Line 151- Please add spp. after Lactobacillus.

R: Fixed.

Line 151- Please replace "Gardnerella vaginalis" by "Gardnerella/Bacteroides spp."

R: Fixed.

Line 152- Please delete "s" in "spps.".

R: Fixed.

Line 152- Why Gram-positive cocci? Please reviewe Nugent score: https://www.ncbi.nlm.nih.gov/pmc/articles/PMC269757/

R: We deleted the Gram positive cocci accourdingly.

Line 156- Why intermediate BV in Nugent score between 4 and 6? It is a intermediate vaginal microbiota that did not achieve a vaginal dysbiosis yet. It is incorrect to cite as intermediate BV.

R: Fixed.

Lines 162-163- Please a further explaination of the AV score is needed for the Readers to clarify the Av diagnosis.

R: As you know each test including AV score interpretation are described in full detail in several studies previously published. Preferably, we used suitable references here to avoid lengthening the text. All readers can refer to the mentioned references for more information.

Line 164- Please change the title because it is incorrect. The section described the isolation of certain bacteria species from vaginal samples.

R: Fixed.

Lines 171-172- Please change 2 in underscript in "H2S" and to add the reference of the classification of the bacterial isolates through the biochemical evaluation.

R: Fixed.

Line 173- Please avoid the abbreviation in the titles. I recommend to do it in the first sentence of the section (line 174).

R: Fixed.

Line 186- Please replace "Collected quantitative data were coded; processed, edited, and analyzed" by "Collected quantitative data were coded and analyzed".

Line 189- Please replace "(IC)" by "(CI)".

R: Fixed.

RESULTS

Line 195- Please replace "for BV and AV" by "for their vaginal microbiota".

R: Fixed.

Line 203- Please replace "yeast cell infection" by "candidiasis" and "Trichomonas vaginalis" by "trichomoniasis (Trichomonas vaginalis)".

R: Fixed.

Lines 202-210- Please avoid the repetition of "422 study participants" and mantain the results form of presentation. I recommend to cite the percentagen (number of cases positive/total number of cases).

R: We revised all results. Also, we calculated all percent again and see some mistakes that all have been corrected and fixed.

Lines 213-215- Please rephrase the sentence. The prevalences of BV between age categories are similar (21.6, 20.7 and 18.3%) and without statistical differences.

R: This part was revised and condensed according to the editor comments.

Lines 228-230- Please rephrase the sentence. The prevalences of BV are similar and without statistical differences.

R: This part was revised and condensed according to the editor comments.

Line 241- Please add "0" in "P = .042".

R: Fixed.

Lines 244-249- Please explain better the results and rectify "P = .042".

R: Fixed.

Lines 250-251- Please explain better. I am a little confused... the prevalence of BV in symptomatic pregnant women was 31.8% and the prevalence of BV in asymptomatic women was 68.2% in Table 4.

R: This part was revised, the table 4 was deleted according to editor recommendation. The results were calculated again and have been writed more clearly (page 19, lines 219-227).

Lines 260-265- Please rephrase the sentences.

R: Fixed.

Lines 273-275- Please rephrase the sentence and rectify "[0.292 (0.102, 0.833)], (P = 0.021))".

R: Fixed.

Line 274- What do you mean with "may significantly associate"?

R: Fixed.

Lines 279-281- Please rephrase the sentence.

 R:This part was revised, the table 6 was deleted according to editor recommendation. The results were calculated again and have been writed more clearly (page 19, lines 219-227).

Line 283- Please clarify to the readers that 44 bacterial isolates are from AV women. BV and AV are both vaginal dysbiosis.

R: Fixed.

Lines 285 until the remaining manuscript- The abreviations GPB (Gram PositiveBacteria) and GNB (Gram Negative Bacteria) were added directly without the full names, but most important they are overgeneralize terms misleading the Readers. All GPB are Staphylococcus genus and all GNP are genera of Enterobacteriaceae family. Please rectify these terms in all manuscript.

R: Fixed.

Lines 288, 293, and 299- I recommend to delete these titles because you did not possess so many results and these 3 paragraphs can be add after the only paragrah after the title "Bacterial isolates among pregnant women with aerobic vaginitis".

R: Fixed.

Lines 302-304- "... including Citrobacter spp. and K. pneumoniae were 100.0% multidrug-resistant (MDR)". This sentence are misleading the Readers. There are only 2 isolates of Citrobacter spp. and just one K. pneumoniae showing also sensitive pattern to antibiotics. Please clarify better this information to the Readers and I recommend to do it also in Staphylococcus isolates.

R: Fixed.

DISCUSSION

Line 309 and the remaining discussion- Please add also another study with BV and AV in Latin America women, more exactly: Ecuador - Salinas et al (2020) https://www.nature.com/articles/s41598-020-74655-z

R: We add this reference in the mentioned place and used it as one of our references.

Lines 312-314- Please put each individual reference to each country.

 R:This part was revised and condensed according to the editor comments.

Lines 333-335- I would invite the authors to further discuss the discrepancies between studies, besides the diagnostic methods. It seems too vague.

R:This part was revised and condensed according to the editor comments. We add some sentences about the BV diagnosis method accordingly.

Lines 348-350- I would invite the authors to further discuss the statistical significance of the results. It is too vague.

R: Fixed.

Lines 356-358- Why it is relevant to the discussion? It seems too vague.

R:This part was revised and condensed according to the editor comments. We add some sentences about the BV diagnoss method.

Lines 368-370- I would invite the authors to further discuss the discrepancies between studies. It seems too vague.

R:This part was revised and condensed according to the editor comments. We add some sentences about the BV diagnoss method.

Lines 382- I would invite the authors to also compare the prevalence of AV in age with Salinas et al (2020) and Pacha-Herrera et al. (2020):

https://www.nature.com/articles/s41598-020-74655-z

https://doi.org/10.3389/fcimb.2020.00303

R:This part was revised and condensed according to the editor comments. However, we made the comparison of our results with the 2 mentioned studies.

Lines 397-398- I would invite the authors to further discuss the differences in lifestyle between these population subsets. It seems too vague.

R:We deleted this vague statement.

Lines 398-400- Please compare the results of Han et al. (2019) with your results.

R: Fixed.

Lines 401-436- Please rewrite this subsection of discussion. There no bacterial profile done in present study, GPB and GNB terms are misleading the Readers.

R: We corrected all the statements regarding GPB,GNB, and bacterial profiles accordingly.

Lines 413-415- Please state the prevalence of S. agalactiae of these studies.

R: Fixed.

CONCLUSION

Lines 439-440- I am a little confused. In table 4, the prevalence of BV was higher in asymptomatic women (68.2%) than in symptomatic women (31.8%) and contradicting the lines 449 and 450 of the Conclusion.

R: We corrected and revised all the results. There was some errors in the calculated values. All corrections were reflected in te results.

I suggest the authors to rewrite the Conclusion section and to add the limitations of the study, which there are several as previously referred in the GENERAL COMMENTS.

R: Fixed.

TABLES

There are too many tables in the manuscript (10 tables) and the authors can easily reduce the number of tables.

R: Fixed.

Tables 1 and 2 can easily become one table. I recommend the authors to see the Table 1 of Salinas et al. (2020) and adapt their Tables 1 and 2 in that format (https://www.nature.com/articles/s41598-020-74655-z/tables/1).

Table 2 did not show intermediate vaginal microbiota.

Table 4 showed "no" instead of "n", as previously referred in other tables.

Table 6 showed "aerobic vaginosis" instead of "aerobic vaginitis".

Table 7 showed information that is repeated in table 10.

Tables 8, 9 and 10 could be combined in one unique table.

General remarks of the tables: * = Significant association can be improve by indicating the statistical analysis made (such as Table 4). Also, the page numbers are wrong after page 30 of the manuscript.

R: We deleted the Tables 4,6, and 7 and prefer to remain the ther tables in the manuscript.

The Table 2 only showed the different infections, and as the intermediate microbiota is not infection, its does not shown. The p-value of * = Significant association is mentioned in each appropriate table.

---

## [Decision Letter · Decision Letter 1]

3 Dec 2021

PONE-D-21-28305R1Prevalence of bacterial vaginosis and aerobic vaginitis and their associated risk factors among pregnant women from northern Ethiopia: a cross-sectional studyPLOS ONE

Dear Dr. Saki,

Thank you for submitting your manuscript to PLOS ONE. After careful consideration, we feel that it has merit but does not fully meet PLOS ONE’s publication criteria as it currently stands. Therefore, we invite you to submit a revised version of the manuscript that addresses the points raised during the review process.

The abstract states: “BV was significantly associated with symptoms of white homogeneous vaginal discharge (P 0.001) and second trimester (P = 0.042). However, AV was significantly associated with secondary school (P = 0.021) and housewife status (P = 0.013).” It would be helpful to include the reference category, i.e: BV was more common in symptomatic vs. asymptomatic people, and in second trimester vs. first trimester samples.

Line 102-103 state: “…numerous research have looked at the link between AV and pregnancy outcomes..” As noted previously, I think this is a mischaracterization. There are very few studies, actually, especially when compared to BV.

Line 115 states that groups have a higher predisposition for BV “…because they have fewer lactobacilli that produce hydrogen peroxide.” I would delete this clause because the causal link between the described features and lower lactobacilli and BV is not proven. It is all associations.

Line 311-314: states “…even higher prevalence was reported among pregnant women who douche their vagina frequently using water (8.9%), those who didn’t use soap during douching (8.4%)”  Both clauses need to describe to what they are compared – the table implies that everyone douched at least once daily – is this true? How is douching characterized?  The second clause also implies that everyone is douching – it would be better represented by use of soap within those who douche. If what is meant is vaginal cleansing with fingers or a cloth, this should be specified. Douching means irrigating the vagina with a liquid.  

Line 515-524: The limitations paragraph should go before the Conclusion section. Additionally, please break out the list of limitations into a more readable set of sentences.

Minor typographical/grammatical errors:

Line 90-91: “However, both AV and BV are vaginal dysbiosis that characterized by the lactobacilli reduction”  - dysbiosis needs to be plural and either the “that” removed, or an “are” added after it. Also, “the lactobacilli reduction” should be “the reduction in lactobacilli.”

Line 97: lactobacillus should be italicized and capitalized

Line 99: “These enzymes sialic stimulate the”. Please remove the word “sialic” here

Line 188: “…Lactobacillus morphotypes i.e. that no morphotypes = 4.” Please remove the word “that”

Line 233-234 states “…study participants were secondary school…”. Do you mean that participants had completed secondary school? Or had some secondary school education? Please specify.

Line 255 states “…results were not significantly associated (P = 0.549).” – please add what results were not associated with (I think it is symptoms, but to make it unambiguous this should be specified).

Lines 285-287: “However, pregnant women with the second trimester had a higher prevalence of BV (23.7%) than those who were in the first and third trimester with a prevalence of (17.6%) and (16.5%), respectively. Pregnant women in the second trimester, had significantly higher BV (P = 0.042).” – In the first sentence, it should be be “…pregnant women sampled in the second…”. In the second sentence, please specify higher in comparison to…(I believe it is those sampled in the third trimester, but again, this should be explicitly stated). (Same comments for line 314-317, however, here you state that the people in second trimester had higher prevalence than those in both first and third trimester, but it appears that the reference group is third trimester only).(Also line 413-414)

Line 304 “…than 21-29 years and those lower than 20 years age…” Please include “women aged” after than, and change the latter part to “20 years of age”

Line 311: pantyliner should be plural

Line 316-317: “…who were got pregnant for the first time…” Please remove the word got

Line 318: “…who were pregnant many times” Please change to “who had been pregnant before…”

Line 318 states that “…but they were not significantly associated…” – to what does “they” refer? Do you mean that gravidity was not associated with AV? If so, please state that.

Line 320: “However, after adjustment of confounders in multivariate analyses, secondary school [0.292 (0.102, 0.833)], P = 0.021] and housewife [2.856 (1.250, 6.523), P = 0.013] pregnant women remained independently associated with a decreased (preventive) and increased likelihood of AV positive, respectively. Please specify reference categories.

Line 333-334 states: “different Staphylococcus strains and Enterobacteriaceae species,” – I believe you mean species for both. A strain is a specific isolate of a given species, and the next sentence talks about different Staph species.  I do think that the term strains is used appropriately in Table 5.

Line 395 states “…the uncultivated bacterial vaginosis-associated bacterium.” This should be “……the uncultivated bacterial vaginosis-associated bacterium-1 (BVAB1).”  (there are three: BVAB1, BVAB2 and BVAB3, which is now known as Mageebacillus indolicus, which is why it is important to specify).

Line 419: “In contrast to the current study, a study reported from South Africa, revealed that HIV-positive item was significantly associated with BV”. Please change the “that HIV-positive status…”

Line 434” “…knowledge, the AV and its associated”. Please delete “the”

Line 436-438: “In this study, no one of socio-demographic and sexual and behavioral characteristics variables was significantly associated with the prevalence of AV in pregnant women except for education and occupation status that was in contrast…” Please delete “one of” and change “was” to “were”

Line 455: Please add “A” at the beginning of this sentence: “Previous study from China reported a higher prevalence rate of mixed AV cases with other infections…”

Line 476: “Also, the gentamicin with susceptibility rate of 92.9% was the third effective antibiotic against Enterobacteriaceae.” Please change to “Also, gentamicin with a susceptibility rate of 92.9% was the third most effective…”

Line 506-512: “The prevalence of BV was higher among symptomatic (35.1%) than asymptomatic pregnant women (16.8%). White vaginal discharge and second trimester were the risk factor and protective item for BV, respectively. Whereas, secondary school and housewife were also found protective and risk factor for AV, respectively. The Staphylococcus spp. were more prevalent than the Enterobacteriaceae in AV pregnant women. Amoxicillin/clavulanate had the highest overall resistance rate against Enterobacteriaceae, while the overall resistance rate of Staphylococcus spp.was observed against penicillin.” – This simply restates what has already been discussed and could be deleted.

We look forward to receiving your revised manuscript.

Kind regards,

Caroline Mitchell

Academic Editor

PLOS ONE

Journal Requirements:

Reviewers' comments:

Reviewer's Responses to Questions

**Comments to the Author**

1. If the authors have adequately addressed your comments raised in a previous round of review and you feel that this manuscript is now acceptable for publication, you may indicate that here to bypass the “Comments to the Author” section, enter your conflict of interest statement in the “Confidential to Editor” section, and submit your "Accept" recommendation.

Reviewer #1: All comments have been addressed

2. Is the manuscript technically sound, and do the data support the conclusions?

Reviewer #1: Yes

3. Has the statistical analysis been performed appropriately and rigorously? 

Reviewer #1: N/A

4. Have the authors made all data underlying the findings in their manuscript fully available?

Reviewer #1: Yes

5. Is the manuscript presented in an intelligible fashion and written in standard English?

Reviewer #1: Yes

6. Review Comments to the Author

Reviewer #1: Congratulations to the authors for the present study.

I recommend your manuscript for publication. However, there are still some minor English editing that should be check by the authors before publication.

Just some examples of minor errors in the revised manuscript with track changes:

Line 446: "In line with the current study, Hassan et al. [11] from Egypt, showed that..."

Line 451: "In line with the current study, Salinas et al. [50] and Pacha-Herrera et al. [76] from Ecuador, did"

But most important all Reviewer's comments have been answered by the authors.

Congratulations and best regards

7. PLOS authors have the option to publish the peer review history of their article (what does this mean?). If published, this will include your full peer review and any attached files.

Reviewer #1: **Yes: **António Machado

---

## [Author Response · Author response to Decision Letter 1]

21 Dec 2021

Dear Dr. Caroline Mitchell 

Academic Editor of PLOS ONE

Subject : Submission of revised manuscript entitled "Prevalence of bacterial vaginosis and aerobic vaginitis and their associated risk factors among pregnant women from northern Ethiopia: a cross-sectional study" (ID: PONE-D-21-28305R1).

Thank you for your email of 3 December, 2021 enclosing the editor and reviewers ʼcomments.We also greatly appreciate the reviewers for their complimentary comments and suggestions. We have carefully reviewed the comments and have revised the manuscript accordingly. Our point to point responses are given below. (The reviewer’s comments are in italics). Changes to the manuscript are indicated in yellow highlighted sentences/words. We hope that you find our responses satisfactory and that the manuscript is now acceptable for publication. Anyway, we should be grateful if you let us know about our further changes required.

Yours sincerely,

Corresponding author: Morteza Saki

Mailing address: Department of Microbiology, Faculty of Medicine, Ahvaz Jundishapur University of Medical Sciences, Ahvaz, Iran

P.O. Box: 159 

Postal Code: 61357-15794 

Email: mortezasaki1981@gmail.com

The followings are point-by-point responses :

The abstract states: “BV was significantly associated with symptoms of white homogeneous vaginal discharge (P 0.001) and second trimester (P = 0.042). However, AV was significantly associated with secondary school (P = 0.021) and housewife status (P = 0.013).” It would be helpful to include the reference category, i.e: BV was more common in symptomatic vs. asymptomatic people, and in second trimester vs. first trimester samples.

Response (R): We fixed what you requested but due to limitation of the word count of abstract, we have to delete some part of manuscript accordingly to reach 300 words.

Line 102-103 state: “…numerous research have looked at the link between AV and pregnancy outcomes..” As noted previously, I think this is a mischaracterization. There are very few studies, actually, especially when compared to BV.

Response (R): Fixed.

Line 115 states that groups have a higher predisposition for BV “…because they have fewer lactobacilli that produce hydrogen peroxide.” I would delete this clause because the causal link between the described features and lower lactobacilli and BV is not proven. It is all associations.

Response (R): Fixed.

Line 311-314: states “…even higher prevalence was reported among pregnant women who douche their vagina frequently using water (8.9%), those who didn’t use soap during douching (8.4%)” Both clauses need to describe to what they are compared – the table implies that everyone douched at least once daily – is this true? How is douching characterized? The second clause also implies that everyone is douching – it would be better represented by use of soap within those who douche. If what is meant is vaginal cleansing with fingers or a cloth, this should be specified. Douching means irrigating the vagina with a liquid. 

 Response (R): Fixed.

Line 515-524: The limitations paragraph should go before the Conclusion section. Additionally, please break out the list of limitations into a more readable set of sentences.

Response (R): Fixed.

Minor typographical/grammatical errors:

Line 90-91: “However, both AV and BV are vaginal dysbiosis that characterized by the lactobacilli reduction” - dysbiosis needs to be plural and either the “that” removed, or an “are” added after it. Also, “the lactobacilli reduction” should be “the reduction in lactobacilli.”

Response (R): Fixed.

Line 97: lactobacillus should be italicized and capitalized

Response (R): Fixed.

Line 99: “These enzymes sialic stimulate the”. Please remove the word “sialic” here

Response (R): Fixed.

Line 188: “…Lactobacillus morphotypes i.e. that no morphotypes = 4.” Please remove the word “that”

Response (R): Fixed.

Line 233-234 states “…study participants were secondary school…”. Do you mean that participants had completed secondary school? Or had some secondary school education? Please specify.

Response (R): Fixed.

Line 255 states “…results were not significantly associated (P = 0.549).” – please add what results were not associated with (I think it is symptoms, but to make it unambiguous this should be specified).

Response (R): Fixed.

Lines 285-287: “However, pregnant women with the second trimester had a higher prevalence of BV (23.7%) than those who were in the first and third trimester with a prevalence of (17.6%) and (16.5%), respectively. Pregnant women in the second trimester, had significantly higher BV (P = 0.042).” – In the first sentence, it should be be “…pregnant women sampled in the second…”. In the second sentence, please specify higher in comparison to…(I believe it is those sampled in the third trimester, but again, this should be explicitly stated). (Same comments for line 314-317, however, here you state that the people in second trimester had higher prevalence than those in both first and third trimester, but it appears that the reference group is third trimester only).(Also line 413-414)

Response (R): Fixed.

Line 304 “…than 21-29 years and those lower than 20 years age…” Please include “women aged” after than, and change the latter part to “20 years of age”

Response (R): Fixed.

Line 311: pantyliner should be plural

Response (R): Fixed.

Line 316-317: “…who were got pregnant for the first time…” Please remove the word got

Response (R): Fixed.

Line 318: “…who were pregnant many times” Please change to “who had been pregnant before…”

Response (R): Fixed.

Line 318 states that “…but they were not significantly associated…” – to what does “they” refer? Do you mean that gravidity was not associated with AV? If so, please state that.

Response (R): Fixed.

Line 320: “However, after adjustment of confounders in multivariate analyses, secondary school [0.292 (0.102, 0.833)], P = 0.021] and housewife [2.856 (1.250, 6.523), P = 0.013] pregnant women remained independently associated with a decreased (preventive) and increased likelihood of AV positive, respectively. Please specify reference categories.

Response (R): Fixed.

Line 333-334 states: “different Staphylococcus strains and Enterobacteriaceae species,” – I believe you mean species for both. A strain is a specific isolate of a given species, and the next sentence talks about different Staph species. I do think that the term strains is used appropriately in Table 5.

Response (R): Fixed.

Line 395 states “…the uncultivated bacterial vaginosis-associated bacterium.” This should be “……the uncultivated bacterial vaginosis-associated bacterium-1 (BVAB1).” (there are three: BVAB1, BVAB2 and BVAB3, which is now known as Mageebacillus indolicus, which is why it is important to specify).

Response (R): Fixed.

Line 419: “In contrast to the current study, a study reported from South Africa, revealed that HIV-positive item was significantly associated with BV”. Please change the “that HIV-positive status…”

Response (R): Fixed.

Line 434” “…knowledge, the AV and its associated”. Please delete “the”

Response (R): Fixed.

Line 436-438: “In this study, no one of socio-demographic and sexual and behavioral characteristics variables was significantly associated with the prevalence of AV in pregnant women except for education and occupation status that was in contrast…” Please delete “one of” and change “was” to “were”

Response (R): Fixed.

Line 455: Please add “A” at the beginning of this sentence: “Previous study from China reported a higher prevalence rate of mixed AV cases with other infections…”

Response (R): Fixed.

Line 476: “Also, the gentamicin with susceptibility rate of 92.9% was the third effective antibiotic against Enterobacteriaceae.” Please change to “Also, gentamicin with a susceptibility rate of 92.9% was the third most effective…”

Response (R): Fixed.

Line 506-512: “The prevalence of BV was higher among symptomatic (35.1%) than asymptomatic pregnant women (16.8%). White vaginal discharge and second trimester were the risk factor and protective item for BV, respectively. Whereas, secondary school and housewife were also found protective and risk factor for AV, respectively. The Staphylococcus spp. were more prevalent than the Enterobacteriaceae in AV pregnant women. Amoxicillin/clavulanate had the highest overall resistance rate against Enterobacteriaceae, while the overall resistance rate of Staphylococcus spp.was observed against penicillin.” – This simply restates what has already been discussed and could be deleted.

Response (R): Thank you for your suggestion. However, Due to the high lenghth of the manuscript, we prefer to remain this summery of the study.

Reviewer #1: 

Just some examples of minor errors in the revised manuscript with track changes:

Line 446: "In line with the current study, Hassan et al. [11] from Egypt, showed that..."

Line 451: "In line with the current study, Salinas et al. [50] and Pacha-Herrera et al. [76] from Ecuador, did"

Response (R): Fixed.

---

## [Editor Report · Decision Letter 2]

4 Jan 2022

Prevalence of bacterial vaginosis and aerobic vaginitis and their associated risk factors among pregnant women from northern Ethiopia: a cross-sectional study

PONE-D-21-28305R2

Dear Dr. Saki,

We’re pleased to inform you that your manuscript has been judged scientifically suitable for publication and will be formally accepted for publication once it meets all outstanding technical requirements.

Kind regards,

Caroline Mitchell

Academic Editor

PLOS ONE
---

## [Editor Report · Acceptance letter]

17 Feb 2022

PONE-D-21-28305R2 

Prevalence of bacterial vaginosis and aerobic vaginitis and their associated risk factors among pregnant women from northern Ethiopia: a cross-sectional study 

Dear Dr. Saki:

I'm pleased to inform you that your manuscript has been deemed suitable for publication in PLOS ONE. Congratulations! Your manuscript is now with our production department. 

Kind regards, 

on behalf of

Dr. Caroline Mitchell 

Academic Editor

PLOS ONE